# Genetically engineered mice for combinatorial cardiovascular optobiology

**Frank K Lee[1], Jane C Lee[1], Bo Shui[1], Shaun Reining[1], Megan Jibilian[1], David M Small[2], Jason S Jones[2], Nathaniel H Allan-Rahill[2], Michael RE Lamont[2], Megan A Rizzo[3], Sendoa Tajada[4], Manuel F Navedo[5], Luis Fernando Santana[4], Nozomi Nishimura[2], Michael I Kotlikoff[1]***

[1]Department of Biomedical Sciences, Cornell University, Ithaca, United States; [2]Department of Biomedical Engineering, Cornell University, Ithaca, United States; [3]Department of Physiology, University of Maryland School of Medicine, Baltimore, United States; [4]Departments of Physiology and Membrane Biology, University of California, Davis School of Medicine, Davis, United States; [5]Department of Pharmacology, University of California, Davis, Davis, United States

**\*For correspondence:**
mik7@cornell.edu

**Abstract** Optogenetic effectors and sensors provide a novel real-time window into complex physiological processes, enabling determination of molecular signaling processes within functioning cellular networks. However, the combination of these optical tools in mice is made practical by construction of genetic lines that are optically compatible and genetically tractable. We present a new toolbox of 21 mouse lines with lineage-specific expression of optogenetic effectors and sensors for direct biallelic combination, avoiding the multiallelic requirement of Cre recombinase -mediated DNA recombination, focusing on models relevant for cardiovascular biology. Optogenetic effectors (11 lines) or $Ca^{2+}$ sensors (10 lines) were selectively expressed in cardiac pacemaker cells, cardiomyocytes, vascular endothelial and smooth muscle cells, alveolar epithelial cells, lymphocytes, glia, and other cell types. Optogenetic effector and sensor function was demonstrated in numerous tissues. Arterial/arteriolar tone was modulated by optical activation of the second messengers $InSP_3$ (optoα1AR) and cAMP (optoß2AR), or $Ca^{2+}$-permeant membrane channels (CatCh2) in smooth muscle (*Acta2*) and endothelium (*Cdh5*). Cardiac activation was separately controlled through activation of nodal/conducting cells or cardiac myocytes. We demonstrate combined effector and sensor function in biallelic mouse crosses: optical cardiac pacing and simultaneous cardiomyocyte $Ca^{2+}$ imaging in $Hcn4^{BAC}$-CatCh2/*Myh6*-GCaMP8 crosses. These experiments highlight the potential of these mice to explore cellular signaling in vivo, in complex tissue networks.

## Introduction

Optogenetic effectors and sensors enable the interrogation of complex biological signaling networks at the molecular level in vivo (*Akerboom et al., 2013*; *Fenno et al., 2011*; *Gengyo-Ando et al., 2017*; *Kotlikoff, 2007*; *Tsien, 2003*). The power of these molecular tools is optimized in mammalian systems by genetic lineage specification and combinatorial strategies. The former provides the ability to activate or determine the responses of individual cellular components of a complex network, whereas the latter, using spectrally compatible optogenetic tools, enables the dissection of the network by activation of defined elements and simultaneous interrogation of individual system component responses, thereby significantly enhancing the ability to understand functional networks. Although simple in concept, the usefulness of optogenetic tools in mammalian systems requires a parsimonious allelic strategy that minimizes experimental crosses, while supporting robust and consistent sensor/effector expression. For example, while DNA recombination strategies can achieve lineage

specificity by crossbreeding Cre recombinase and floxed responder mouse lines (*Zariwala et al., 2012*), the attendant allelic requirements render this approach impractical for combinatorial experiments using multiple optogenetic tools. Such experiments are facilitated by the design and assembly of sets of individual monoallelic lines that strategically place spectrally distinct optogenetic effectors and sensors in interacting cell lineages, allowing simple crosses to probe cell-cell signaling. Here, we report a marked expansion of the mammalian optogenetic toolbox through the development of multiple mouse lines designed for combinatorial experiments within the cardiovascular system. Interacting vascular, neural, and muscular tissues are targeted through a variety of genetic strategies, expression specificity confirmed, and examples of their use provided.

## Results

We report the generation of CHROMus (Cornell/National Heart Lung Blood Resource for Optogenetic Mouse Signaling) mice, 21 lines of optogenetic sensor and effector mice, in which the expression of the transgene is lineage specified (*Table 1*) and demonstrate the utility of these lines for in vivo and ex vivo imaging. The optogenetic effector proteins we selected for this toolbox were optoα1AR (*Airan et al., 2009*), optoß2AR (*Airan et al., 2009*), and CatCh2 (*Zhang et al., 2006*) for light activation of InsP$_3$ or cAMP secondary messenger pathways or Ca$^{2+}$ membrane channel, respectively. Each of the open reading frames for the optogenetic proteins were linked to an internal ribosome entry sequence (IRES) element and bacterial LacZ ORF for simple screening. The fluorescent GECI proteins, GCaMP and RCaMP, in reporter lines were derived from circularly permutated EGFP and RFP constructs for in vivo, real-time reporting of cellular Ca$^{2+}$ signaling, as shown in *Figure 1—figure supplement 1*.

CHROMus mice can be used directly as individual optogenetic effector or fluorescent reporter lines to study real-time response to light activation of the target tissue or to study cellular or subcellular Ca$^{2+}$ signaling in response to various physiological stimuli. However, the system provides the novel opportunity to create novel effector/reporter mouse combinations to study the complex network of physiological signals and responses, as illustrated in *Figure 1*. Here, activation of the CatCh2 protein expressed in sinoatrial (SA) nodal pacemaker cells triggers cardiac conduction, as monitored by GCaMP8 fluorescence in ventricular myocytes. Other combinatorial preparations from the CHROMus toolbox enable the evaluation of cellular function in mouse lines that simulate human disease. We have constructed 11 mouse lines expressing various optogenetic effector proteins in seven different lineages and 10 expressing fluorescent GECI reporter proteins in seven lineages across a wide range of tissues, organs, and systems (*Table 1*).

### Cardiac conduction system optogenetic mice

The potassium/sodium hyperpolarization-activated cyclic nucleotide-gated channel 4 protein (HCN4) underlies the pacemaker current in the SA node, controlling heart rate and heart rhythm (*DiFrancesco, 1993*; *Moosmang et al., 2001*). For light-activated control of heart rhythm, we constructed a *Hcn4*$^{BAC}$-CatCh2_IRES_lacZ mouse line, where the expression of the optoeffector CatCh2 and a marker protein (LacZ) are under transcriptional control of the *Hcn4* locus in a BAC (*Table 1*), thus directing expression to the cells of the cardiac conductance system. X-gal staining of adult *Hcn4*$^{BAC}$-CatCh2_IRES_LacZ mice demonstrated specific staining in cells of the cardiac conductance system, including the SA node region in the right atrium, the atrioventricular junction, and the ventricular conducting network (*Figure 2A*). LacZ-expressing cells were observed at the base of the right carotid vein where it joins the right atrium (*Figure 2—figure supplement 1A*) and along the right carotid vein to the base of the common carotid vein (*Figure 2—figure supplement 1B*). Expression was also documented in embryonic (E10.5) and neonatal (PN4) hearts in the atrium and the ventricular conducting system (*Figure 2B*).

To confirm optical control of heart rhythm, we stimulated the SA node area with 473 nm light in vivo while recording the ECG. As shown in *Figure 2C*, cardiac conduction was driven with stimulation between 2 Hz and 4 Hz, with dropouts at higher frequencies. Heart rhythm returned to normal following optical stimulation (*Figure 2—figure supplement 1C*), and illumination of off-target tissue (left ventricle) did not result in optical rhythm control (*Figure 2—figure supplement 1D*). Pacing was maintained with laser pulse lengths varying between 15 and 70 ms (data not shown). The increase in laser-stimulated R-wave duration is shown as a widening of the R-wave in the R-wave-centered

**Table 1.** List of BACs and PCR primers for CHROMus mice.

| CHROMus ID | Transgenic line | Tissue/sensor type (effector [E] or sensor [S]) | Promoter or BAC | Genotyping primer 1 | Genotyping primer 2 |
|---|---|---|---|---|---|
| B3 | Acta2$^{BAC}$–RCaMP1.07 | Smooth muscle – S | RP23-370F21 | GCTTGTCTGTAAGCGGATGCC | TGCTGCTGCCACTCTAGTGAGAAA |
| B4 | Myh6-GCaMP8 | Cardiac muscle – S | Addgene 55594 | AAGGGCGAGGAGCTGTTCA | CGATCTGCTCTTCAGTCAGTTGGT |
| B5 | Myh6-CatCh2_IRES_lacZ | Cardiac muscle – E | Addgene 55594 | GGAGATCTATGTGTGCGCTATC | CCAGCAGTTCTTCGACATCA |
| B6 | Acta2$^{BAC}$–Opto α 1AR_IRES_lacZ | Smooth muscle – E | RP23-370F21 | GGGTTGGTCCCGCTATATTC | GAAGGCAGGGATGGTCATAAA |
| B7 | Acta2$^{BAC}$–Optoβ2AR_IRES_lacZ | Smooth muscle – E | RP23-370F21 | ATAGCTCTCAGCAACCTGTTGGGT | TTCAACACGGTTTGGAGGCG |
| B8 | Acta2$^{BAC}$–CatCh2_IRES_lacZ | Smooth muscle – E | RP23-370F21 | GGAGATCTATGTGTGCGCTATC | CCAGCAGTTCTTCGACATCA |
| B10 | Hcn4$^{BAC}$–GCaMP8 | Cardiac conduction – S | RP23-281H22 | AAGGGCGAGGAGCTGTTCA | CGATCTGCTCTTCAGTCAGTTGGT |
| B14 | Acta2$^{BAC}$–GCaMP_GR | Smooth muscle – S | RP23-370F21 | AAGGGCGAGGAGCTGTTCA | CGATCTGCTCTTCAGTCAGTTGGT |
| B15 | Gja5$^{BAC}$–GCaMP_GR | Endothelium – S | RP24-255O4 | AAGGGCGAGGAGCTGTTCA | CGATCTGCTCTTCAGTCAGTTGGT |
| B16 | Dbh$^{BAC}$–CatCh2_IRES_lacZ | Sympathetic – E | RP23-354N13 | GGAGATCTATGTGTGCGCTATC | CCAGCAGTTCTTCGACATCA |
| B17 | Hcn4$^{BAC}$–CatCh2_IRES_lacZ | Cardiac conduction – E | RP23-281H22 | GGAGATCTATGTGTGCGCTATC | CCAGCAGTTCTTCGACATCA |
| B19 | Acta2$^{BAC}$–GCaMP2 | Smooth muscle – S | RP23-370F21 | AAGGGCGAGGAGCTGTTCA | CGATCTGCTCTTCAGTCAGTTGGT |
| B20 | Cdh5$^{BAC}$–GCaMP8 | Endothelium – S | RP23-453P1 | AAGGGCGAGGAGCTGTTCA | CGATCTGCTCTTCAGTCAGTTGGT |
| B22 | Lck$^{BAC}$–Opto α 1AR_IRES_lacZ | T cells – E | RP24-159E19 | GGGTTGGTCCCGCTATATTC | GAAGGCAGGGATGGTCATAAA |
| B23 | Sftpc$^{BAC}$–GCaMP8 | Alveolar – S | RP23-247J9 | AAGGGCGAGGAGCTGTTCA | CGATCTGCTCTTCAGTCAGTTGGT |
| B26 | Cdh5$^{BAC}$–Opto α 1AR_IRES_lacZ | Endothelium – E | RP23-453P1 | GGGTTGGTCCCGCTATATTC | GAAGGCAGGGATGGTCATAAA |
| B27 | Cdh5$^{BAC}$–Optoβ2AR_IRES_lacZ | Endothelium – E | RP23-453P1 | AACCTTGGAGGCTGGAAAGTAG | TTCAACACGGTTTGGAGGCG |
| B28 | Cdh5$^{BAC}$–CatCh2_IRES_lacZ | Endothelium – E | RP23-453P1 | GGAGATCTATGTGTGCGCTATC | CCAGCAGTTCTTCGACATCA |
| B34 | Acta2$^{BAC}$–GCaMP8.1_mVermilion | Smooth muscle – S | RP23-370F21 | AAGGGAGAGGAGCTGTTCA | CGATCTGCTCCTCGTCTGTCAGCTGGT |
| B35 | Foxj1$^{BAC}$–GCaMP8.1 | Ciliated epithelia – S | RP23-294J17 | AAGGGAGAGGAGCTGTTCA | CGATCTGCTCCTCGTCTGTCAGCTGGT |
| B36 | Aldh1l1$^{BAC}$–Opto α 1AR_IRES_lacZ | Glia – E | RP23-7M9 | GGGTTGGTCCCGCTATATTC | GAAGGCAGGGATGGTCATAAA |

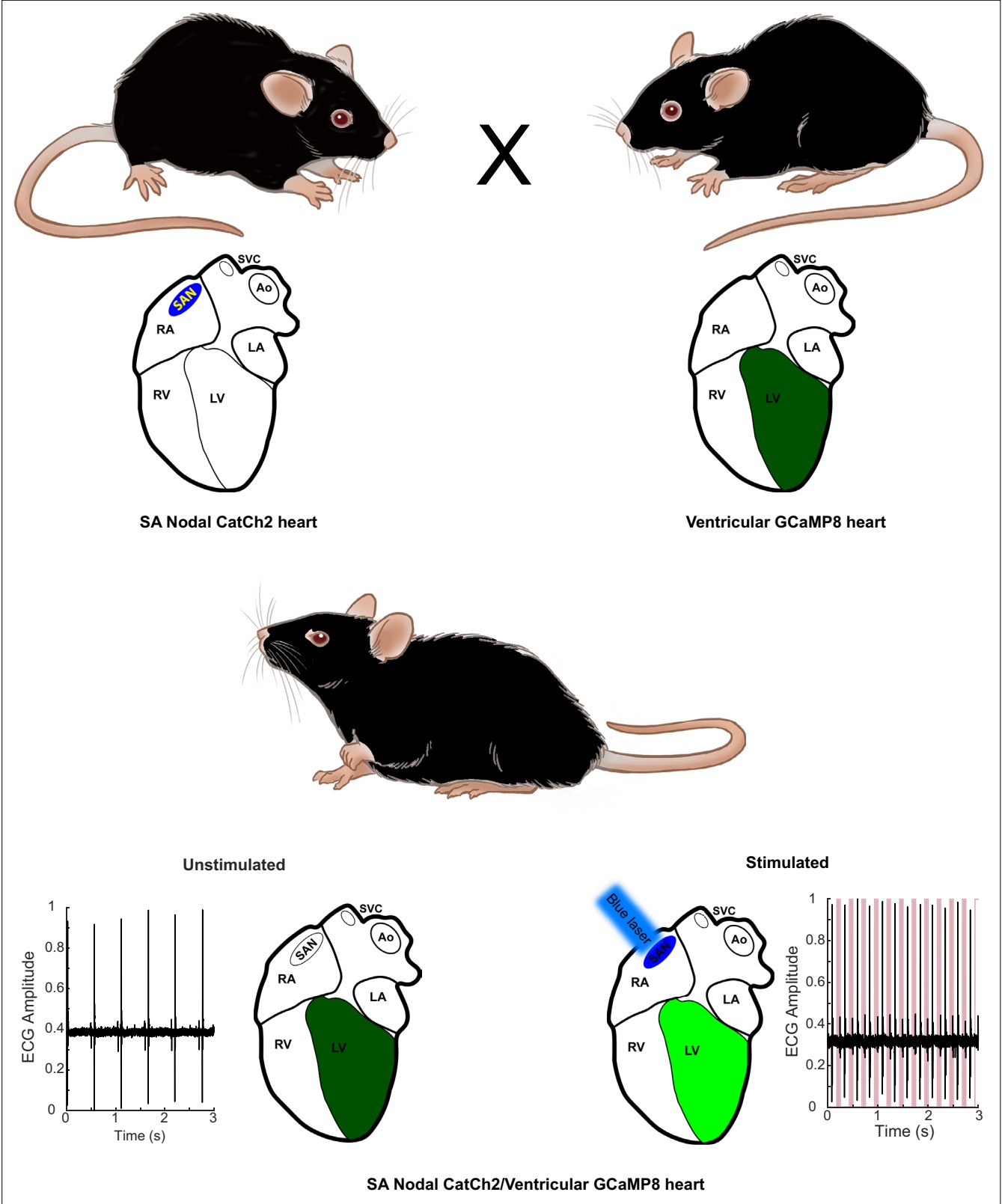

**Figure 1.** Combinatorial optogenetic mice. The figure illustrates the general paradigm of tissue-specific cellular activation and simultaneous response decoding with tissue-specific optical sensors. Here, a biallelic SA nodal CatCh2/cardiomyocyte-GCaMP8 mouse is generated from a single cross of isogenic (C57BL/6J) , hemi-allelic effector/sensor mice, as in *Figure 2H*. Mice with specific expression of optoeffectors (cardiac conducting tissue, cardiac and smooth muscle syncytia, endothelium, sympathetic neurons, T cells, and glia) and optosensors (SA nodal tissue, cardiac and smooth muscle,

*Figure 1 continued on next page*

endothelium, ciliated epithelium, and alveolar epithelium) are reported. LA: left atrium; RA: right atrium; RV: right ventricle; LV: left ventricle; Ao: aorta; SVC: superior vena cava; SAN: sinoatrial node.

The online version of this article includes the following figure supplement(s) for figure 1:

**Figure supplement 1.** Diagram of expression cassettes used in construction of CHROMus mice listed in Table 1.

average of ECG for stimulated and unstimulated beats and the extended PR interval in the P-wave-centered average of ECG sections (*Figure 2D and F*). ECG parameters indicated that R-wave duration and PR interval increased significantly, relative to the autonomously activated cardiac cycle (*Figure 2E and G*), likely indicating a more peripheral activation of SA nodal conducting tissue.

To demonstrate the feasibility of combining optogenetic effector and sensor lines, we crossed *Hcn4*BAC-CatCh2_IRES_LacZ and *Myh6*-GCaMP8 mice. Pacing by laser stimulation resulted in coupling between optical stimulation and GCaMP8 fluorescence, as shown in *Figure 2H* and *Figure 2—video 1*. Differences in effector and sensor kinetics must be considered at high stimulation rates, as previously reported (*Tallini et al., 2006a*). *Hcn4*BAC-CatCh2 mice could also be mated to *Acta2*BAC-GCaMP_GR (Figure 4) and *Cdh5*BAC-GCaMP8 lines (Figure 6) to study smooth muscle or endothelial cell responses to the cardiac pacing. GCaMP_GR (for green/red) is a ratiometric fusion protein consisting of an enhanced GCaMP sequence fused to mCherry (*Shui et al., 2014*).

For the study of the development of the cardiac conduction system and disturbances of conduction in disease states, we developed a *Hcn4*BAC-GCaMP8 mouse line. GCaMP8 is expressed in the *sinus venosus* (*Figure 2I*, left panels) of the embryonic heart and in the SA nodal area at the junction of right atrium and right carotid vein in the adult heart, extending down to the base of the common carotid artery (*Figure 2I*, right panel), similar to the expression pattern in *Hcn4*BAC-CatCh2 mice (*Figure 2— figure supplement 1B*). In developing embryos, spontaneous GCaMP8 signaling was observed in the heart at embryonic day (E) 8.5, E10.5, and E13.5, as well as at postnatal day (PN) 4 (*Figure 2— figure supplement 2A*). Conduction of the GCaMP8 fluorescence signal was observed moving from the atrium to the ventricle in the E13.5 embryo (*Figure 2—video 2*). In the adult heart, GCaMP8 signaling was observed prior to spontaneous ventricular contractions (*Figure 2—figure supplement 2B*, *Figure 2—video 3*).

## Cardiac myocyte optogenetic mice

We constructed two optogenetic lines designed to study $Ca^{2+}$ signaling and electrical communication between heart cells using a minimal promoter for MYH6 protein (*Gulick et al., 1991*). In order to optically initiate cardiac dysrhythmias at virtually any surface region of the heart, we developed the *Myh6*-CatCh2_IRES_lacZ strain, which expresses the CatCh2 and LacZ transgenes in cardiomyocytes. X-gal-stained cryosections of the heart showed widespread staining of cardiomyocytes throughout the heart (*Figure 3A*). Laser irradiation (473 nm) of the left ventricle enabled the stimulation of ventricular premature activations at lower frequencies and complete ventricular pacing at 8 Hz; above 8 Hz, pacing was lost but irregular ventricular initiation could be produced, consistent with the refractory period of ventricular myocytes (*Figure 3B*). As expected, laser stimulation of the left ventricle from a control wildtype mouse did not result in pacing (*Figure 3—figure supplement 1*). As predicted for ventricular activation, the ECG showed markedly enhanced QRS duration without corresponding P-waves, as visualized in an R-wave-centered average of ECG sections for stimulated and unstimulated beats in three mice (*Figure 3C*). Differences in ECG morphology among animals at baseline, as well as unsuccessful stimulation for some beats, account for high variability and abnormal morphology in the averaged trace. The mean R-wave duration significantly increased upon laser stimulation compared to the native unstimulated heartbeat in all three mice examined (*Figure 3D*). Higher magnification ECG traces of autonomous heart activation following ventricular laser stimulation show typical widened QRS complexes (red) followed by return to normal heart activation (*Figure 3E and F*).

We also created the *Myh6*-GCaMP8 line to enable visualization of in vivo, real-time, cardiac $Ca^{2+}$ responses under different experimental stimuli and disease conditions. Cryosections from the heart revealed widespread GCaMP8 fluorescence and immunoreactivity in hearts from this line (*Figure 3G*).

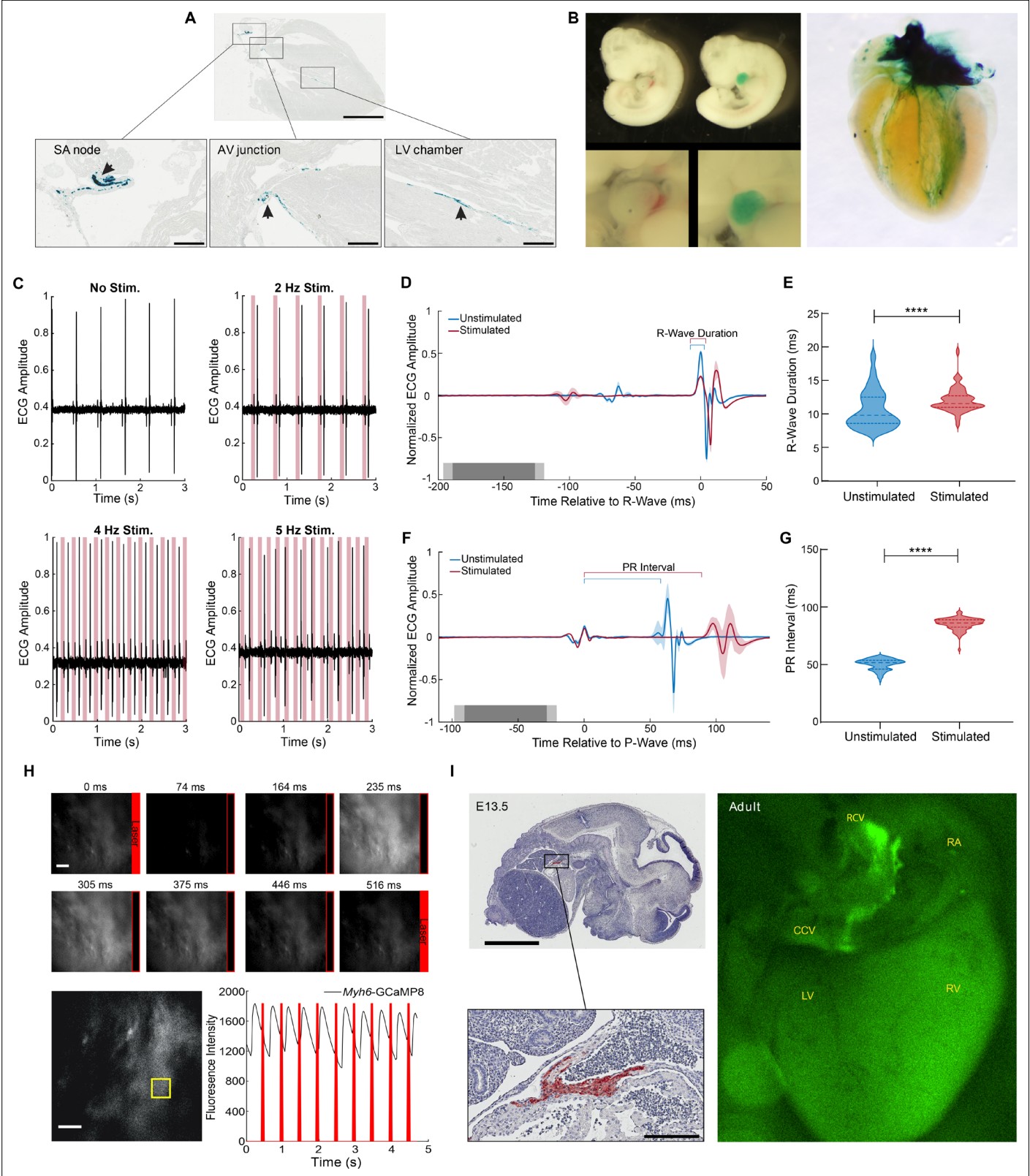

**Figure 2.** Cardiac conduction system optoeffector and optosensor mice. (**A**) X-gal staining of *Hcn4*^BAC-CatCh2_IRES_lacZ adult heart shows LacZ expression in the sinoatrial (SA) node, AV junction, and the ventricular conduction network (arrows). Scale bars: upper panel, 2 mm, lower panels, 300 μm. (**B**) Left panel: X-gal staining of wildtype and *Hcn4*^BAC-CatCh2_IRES_lacZ E10.5 embryos; upper images: whole embryo: lower images: cardiac region. Right panel: whole heart X-gal staining of neonatal heart showing staining in the cardiac conduction system including the SA nodal region,

*Figure 2 continued on next page*

*Figure 2 continued*

atrioventricular junction, and ventricle chambers. (**C**) ECG recording in an anesthetized *Hcn4*[BAC]-CatCh2_IRES_lacZ mouse during intravital pacing with laser illumination. ECG recordings from laser activation of CatCh2 protein demonstrate pacing from 2 to 4 Hz (3 Hz not shown), and pacing maximum. (**D**) ECG averaged by aligning R-wave peaks during laser-stimulated and unstimulated beats (shading indicates standard deviation, n = 273 stimulated beats, n = 115 unstimulated beats from same mouse as in [**E**], dark and light gray indicate average and standard deviation of laser pulse onset and offset times). Laser power is 32.2 mW, pulse length is 70 ms. (**E**) R-wave duration in stimulated and unstimulated beats (****p<0.0001, Mann–Whitney test). Solid lines and dashed lines show median and quartiles. Source data: *Figure 2—source data 1*. (**F**) Averaged ECG aligned by P-wave peak during stimulated and unstimulated beats. (**G**) Quantification of PR interval duration shows elongation in stimulated compared to unstimulated beats (****p<0.0001, Mann–Whitney test). Source data: *Figure 2—source data 2*. (**H**) Fluorescent $Ca^{2+}$ measurement during optical stimulation of SA nodal tissue in *Hcn4*[BAC]-CatCh2_IRES_lacZ /*Myh6*-GCaMP8 perfused mouse heart at 2 Hz shows combination of optogenetic effectors and sensors through optical pacing of the SA node and fluorescent $Ca^{2+}$ responses from ventricular myocytes. Top: fluorescent images from left ventricle with timing of laser pulses indicated by red box. Bottom: plot of GCaMP8 intensity from left ventricle in the outlined box (left image). Source data: *Figure 2—source data 3*. Scale bar: 200 μm. (**I**) Expression of GCaMP8 protein in embryonic and adult hearts from *Hcn4*[BAC]-GCaMP8 mice. Left: anti-GFP immunostaining of paraffin section from E13.5 embryo. Scale bars: 2 mm and 200 μm; right: dorsal view of an adult heart showing native GCaMP8 fluorescence in the SA nodal region, the right carotid vein, and the base of common carotid vein. RA: right atrium; CCV: common carotid vein; RCV: right carotid vein; LV: left ventricle; RV: right ventricle. All images shown are representative images from three animals unless otherwise specified. The ECG data are from a single animal.

The online version of this article includes the following video, source data, and figure supplement(s) for figure 2:

**Source data 1.** R-wave.

**Source data 2.** PR interval.

**Source data 3.** Flourescence.

**Figure supplement 1.** *Hcn4*[BAC]-CatCh2_IRES_LacZ mice.

**Figure supplement 2.** GCaMP8 fluorescence in hearts from *Hcn4*[BAC]-GCaMP8 mice.

**Figure 2—video 1.** GCaMP8 fluorescence from laser pacing of *Hcn4*[BAC]-CatCh2/*Myh6*-GCaMP8 heart.
https://elifesciences.org/articles/67858/figures#fig2video1

**Figure 2—video 2.** GCaMP8 fluorescence in a E13.5 heart from an *Hcn4*[BAC]-GCaMP8 mouse.
https://elifesciences.org/articles/67858/figures#fig2video2

**Figure 2—video 3.** GCaMP8 fluorescence in the atrioventricular junction of *Hcn4*[BAC]-GCaMP8 adult heart.
https://elifesciences.org/articles/67858/figures#fig2video3

To obtain high-resolution intravital images of a beating heart, we injected Texas Red-conjugated dextran retro-orbitally to label the vasculature and acquired two-channel, two-photon images of the left ventricle (*Figure 3H*). GCaMP8 $Ca^{2+}$ fluorescence was observed in cardiomyocytes in the stabilized ventricular free wall during the normal cardiac cycle, coincident with heart contractions. The vascular dextran fluorescence signal reports only motion-related fluorescence changes (*Figure 3I*, *Figure 3—video 1*).

## Smooth muscle optogenetic mice

We constructed optogenetic effector and sensor BAC transgenic mice using the α-Actin2 gene locus, which is selectively expressed in smooth muscle (*Mack and Owens, 1999*). Sensor lines expressing GCaMP_GR protein (modified GCaMP3 fused to mCherry; *Shui et al., 2014*) or RCaMP1.07 (*Ohkura et al., 2012a*) fluorescent $Ca^2$ reporters and effector lines expressing the optoα1AR (*Airan et al., 2009*), optoß2AR (*Airan et al., 2009*), or CatCh2 (*Zhang et al., 2006*) proteins were developed under smooth muscle-specific transcriptional control.

We engineered three recombinant BAC constructs to drive expression of red or green-red sensors in smooth muscle cells (*Figure 1—figure supplement 1*). The *Acta2*[BAC]-RCaMP1.07 (*Bethge et al., 2017*) line expresses the sensor selectively in arterial smooth muscle (*Figure 4—figure supplement 1A*, upper panels). In vivo imaging revealed cyclic fluorescence increases associated with contractions of the right atrium (*Figure 4—figure supplement 1A*, lower panel; *Figure 4—video 1*). When used with optoeffector mice, the line eliminates spectral overlap with excitation of GFP-based sensors.

*Acta2*[BAC]-GCaMP_GR mice provide the advantage of ratiometric $Ca^{2+}$ determination, as well as ease of expression determination in vivo. As shown in brain surface vessel images (*Figure 4A*, upper panel), the GCaMP_GR fusion protein is strongly expressed in the arterial system extending to the arteriolar-capillary junctional cells, but not in the capillary network. To further enhance the intensity of

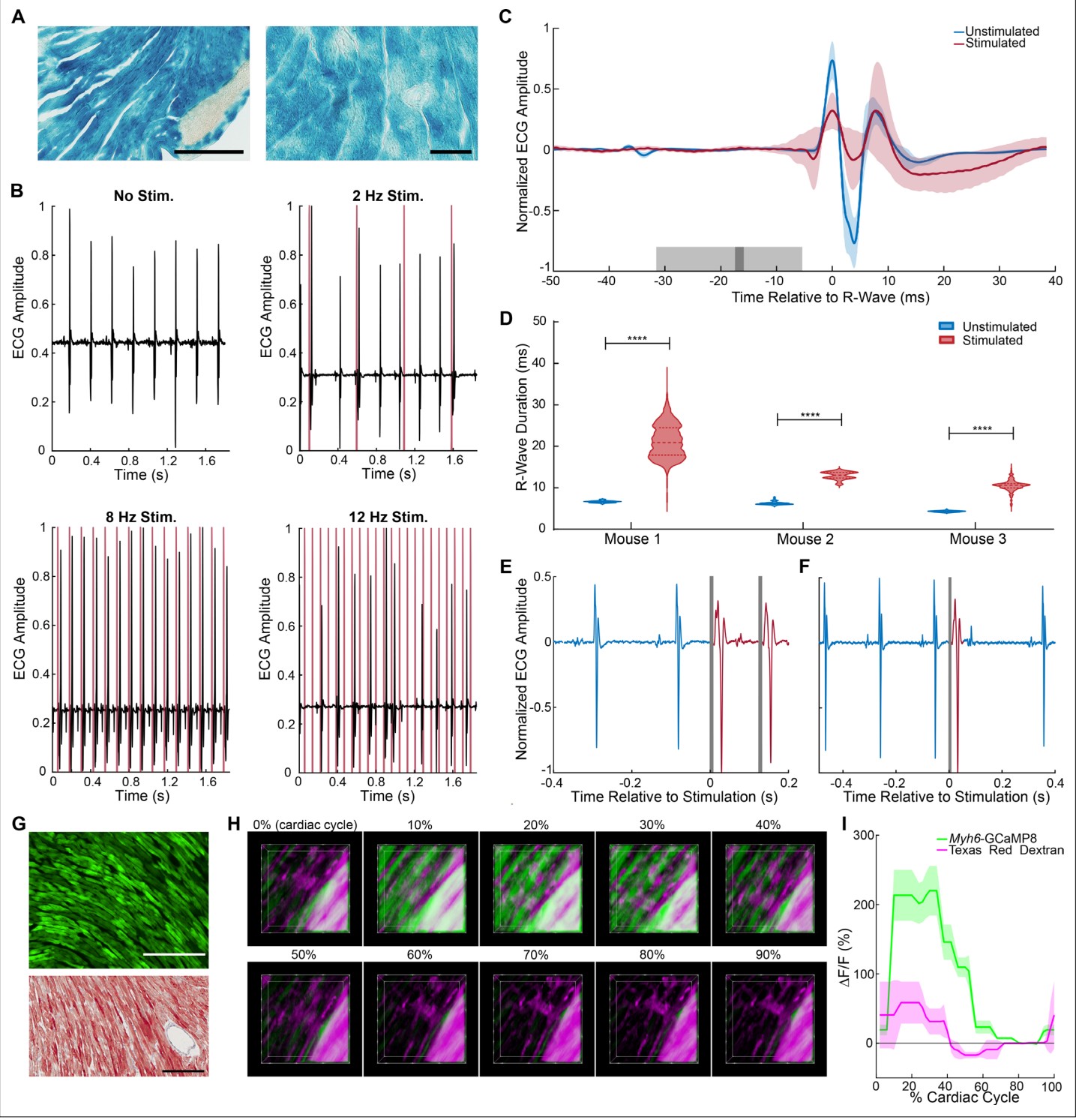

**Figure 3.** Cardiomyocyte optoeffector and optosensor mice. (**A**) X-gal staining of cardiomyocytes in *Myh6*-CatCh2_IRES_lacZ, scale bar: 200 and 60 μm. (**B**) ECG recording during intravital laser stimulation of the CatCh2 protein in the left ventricle at various frequencies (laser power is 32.2 mW, average pulse length is 14 ms). (**C**) R-wave-centered average of all stimulated and unstimulated beats shows R-wave widening (shading indicates standard deviation of ECG signal, dark and light gray indicate average and standard deviation of laser pulse onset and offset times). (**D**) R-wave durations with and without laser stimulation of the transgenic CatCh2 protein are shown from three different mice (****p<0.0001, Mann–Whitney test). Source data: *Figure 3—source data 1* (**E**) ECG recordings demonstrating transition from autonomous to laser-stimulated conduction. (**F**) Premature ventricular contractions evoked by laser stimulation of *Myh6*-CatCh2 ventricular myocytes. (**G**) Expression of GCaMP8 in the heart of an adult *Myh6*-GCaMP8 mouse; top: native fluorescence; bottom: anti-GFP immunohistochemistry; scale bars: 200 μm. (**H**) Intravital two-photon microscopy of the *Myh6*-

Figure 3 continued

GCaMP8 heart. Frames from a three-dimensional reconstruction of *Myh6*-GCaMP8 fluorescence over a cardiac cycle. Each frame represents a 10% increment of the cardiac cycle defined by the peak of the R-wave. The x-axis is the anterior to posterior direction, the y-axis is the apex to base direction, and the z-axis is the epicardium to endocardium direction of the left ventricle free wall; 166 × 166 × 100 μm (xyz) region shown. GCaMP8 fluorescence in cardiomyocytes (green); vasculature labeled with Texas Red-conjugated dextran (magenta). (**I**) Average and standard deviation of change in fluorescence intensity of the imaged region demonstrating Ca$^{2+}$ transients during the cardiac cycle. All images shown are representative images from three animals unless otherwise specified. The ECG data are from three animals.

The online version of this article includes the following video, source data, and figure supplement(s) for figure 3:

**Source data 1.** R-wave duration.

**Figure supplement 1.** Laser stimulation of wildtype left ventricle.

**Figure 3—video 1.** Image stack from two-photon imaging of ventricle contraction in *Myh6*-GCaMP8.
https://elifesciences.org/articles/67858/figures#fig3video1

red fluorescence, we created a second dual-color sensor line using the same transcriptional strategy, but substituting mVermilion, a novel monomeric mCherry variant with a twofold increase in brightness but equivalent excitation and emission profiles. *Acta2*$^{BAC}$-GCaMP8.1_mVermilion mice displayed a similar expression pattern (*Figure 4A*, lower panel). In vivo experiments with these lines indicated robust smooth muscle Ca$^{2+}$-dependent fluorescence. Contractions of blood vessels, airways, and gastrointestinal organs demonstrated increases in green fluorescence during muscle contraction. As shown in *Figure 4—figure supplement 1B* and *Figure 4—video 2*, in vivo fluorescent imaging of spontaneous contractions in the large intestine showed a wave of GCaMP signaling in the longitudinal muscles that was coupled to contractions, whereas mCherry fluorescence remained constant (data not shown).

We constructed three effector lines for smooth muscle optical activation using the *Acta2* BAC (RP23-370F21; *Table 1*) and examined the expression of the LacZ protein located downstream of the optogenetic effector coding sequence in different tissues from *Acta2*$^{BAC}$-Optoα1AR_IRES_lacZ, *Acta2*$^{BAC}$-Optoß2AR_IRES_lacZ and *Acta2*$^{BAC}$-CatCh2_IRES_lacZ mice by X-gal staining. The staining data show widespread expression of all three constructs in the brain arterial system (*Figure 4B*), including the arteriolar/capillary junction. LacZ activity was also observed in arteries supplying the diaphragm, smooth muscles surrounding coronary blood vessels, the bladder, bronchiolar and vascular smooth muscles in lung, and other tissues such as the small intestines, kidney. and uterus (data not shown). X-gal staining of a non-transgenic littermate did not detect LacZ activity (*Figure 4—figure supplement 1C*).

As receptor-mediated activation of phospholipase C or adenylyl cyclase (with associated increases in InsP$_3$ or cAMP) mediates smooth muscle contraction and relaxation, respectively, we examined constriction or dilation of isolated cerebral and mesenteric arteries in response to optical activation of the chimeric Optoα1AR and Optoß2AR proteins. Brief exposure of arteries isolated from *Acta2*$^{BAC}$-Optoα1AR_IRES_lacZ mice to 405 nm light resulted in vasoconstriction (*Figure 5A*, left panel), while light-activated arteries from *Acta2*$^{BAC}$-Optoß2AR_IRES_lacZ lines underwent slow dilations (*Figure 5A*, middle panel; *Figure 5—video 1*), demonstrating functional expression of the proteins and expected physiological responses. The activation of channelrhodopsin in arteries isolated from *Acta2*$^{BAC}$-CatCh2_IRES_lacZ mouse resulted in rapid constriction (*Figure 5A*, right panel; *Figure 5—video 2*). Laser stimulation of arteries from control wildtype mice did not result in vasodilation or vaso constriction similar to the transgenic arteries (*Figure 5—figure supplement 1*). Optoeffector coupling was demonstrated at a cellular level in whole-cell, voltage-clamped arterial myocytes. We first examined coupling through the known linkage between cAMP/protein kinase A signaling and voltage-gated potassium channels in arterial myocytes (*Johnson et al., 2009*; *Luykenaar and Welsh, 2007*; *Nelson and Quayle, 1995*; *Wellman et al., 2001*). Macroscopic I$_K$ currents from single voltage-clamped myocytes isolated from male mesenteric and mid-cerebral arteries were recorded before and after photoactivation of Optoß2AR. I$_K$ (composed of Kv1, Kv2.1, and BK currents) was activated at 10 s intervals by the application of a 500 ms step depolarization from the holding potential of –70 mV to +60 mV. Activation of the Optoß2AR protein by 488 nm light increased the amplitude of I$_K$ 10 s after the exposure, reaching a higher, relatively stable plateau approximately 80 s after photoactivation (*Figure 5B*). Although the dispersion of I$_K$ amplitudes increased about 150 s after activation of Optoß2AR, I$_K$ remained higher than under control conditions for at least 500 s. Exposure to a second

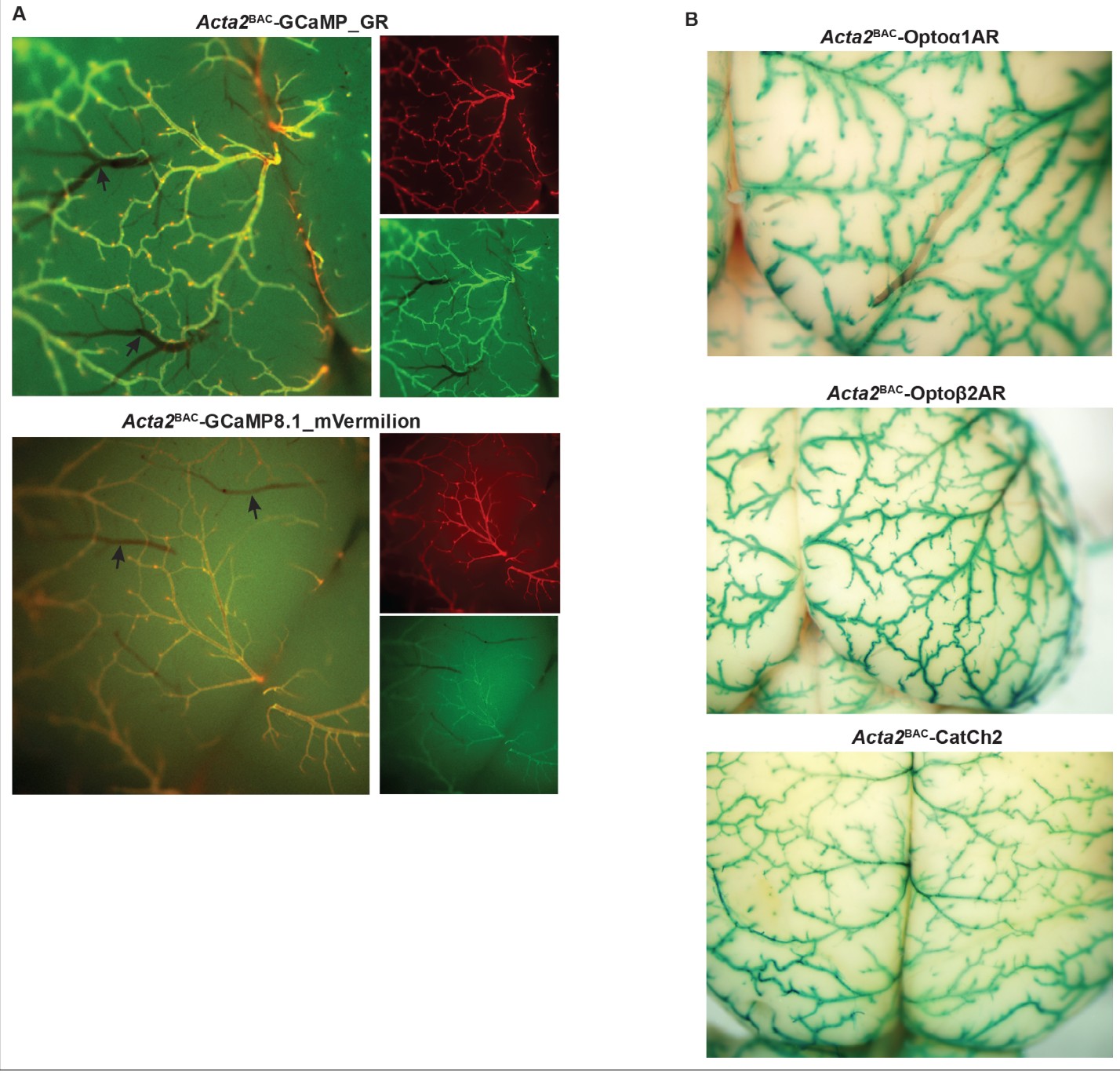

**Figure 4.** Smooth muscle optoeffector and optosensor expression. (**A**) Native fluorescence of surface blood vessels from *Acta2*^BAC^-GCaMP_GR (upper) and *Acta2*^BAC^-GCaMP8.1_mVermilion brain (lower). Note the absence of fluorescence in veins (arrows). Left panel: overlay; right panels: red and green channels. (**B**) X-gal staining of brain surface arteries from *Acta2* optoeffector mice. All images shown are representative images from three animals unless otherwise specified.

The online version of this article includes the following video and figure supplement(s) for figure 4:

**Figure supplement 1.** *Acta2*^BAC^ optogenetic reporter and effector lines.

**Figure 4—video 1.** RCaMP1.07 fluorescence in right atrium of *Acta2*^BAC^-RCaMP1.07.
https://elifesciences.org/articles/67858/figures#fig4video1

**Figure 4—video 2.** GCaMP fluorescence in large intestine of *Acta2*^BAC^-GCaMP_GR mouse.
https://elifesciences.org/articles/67858/figures#fig4video2

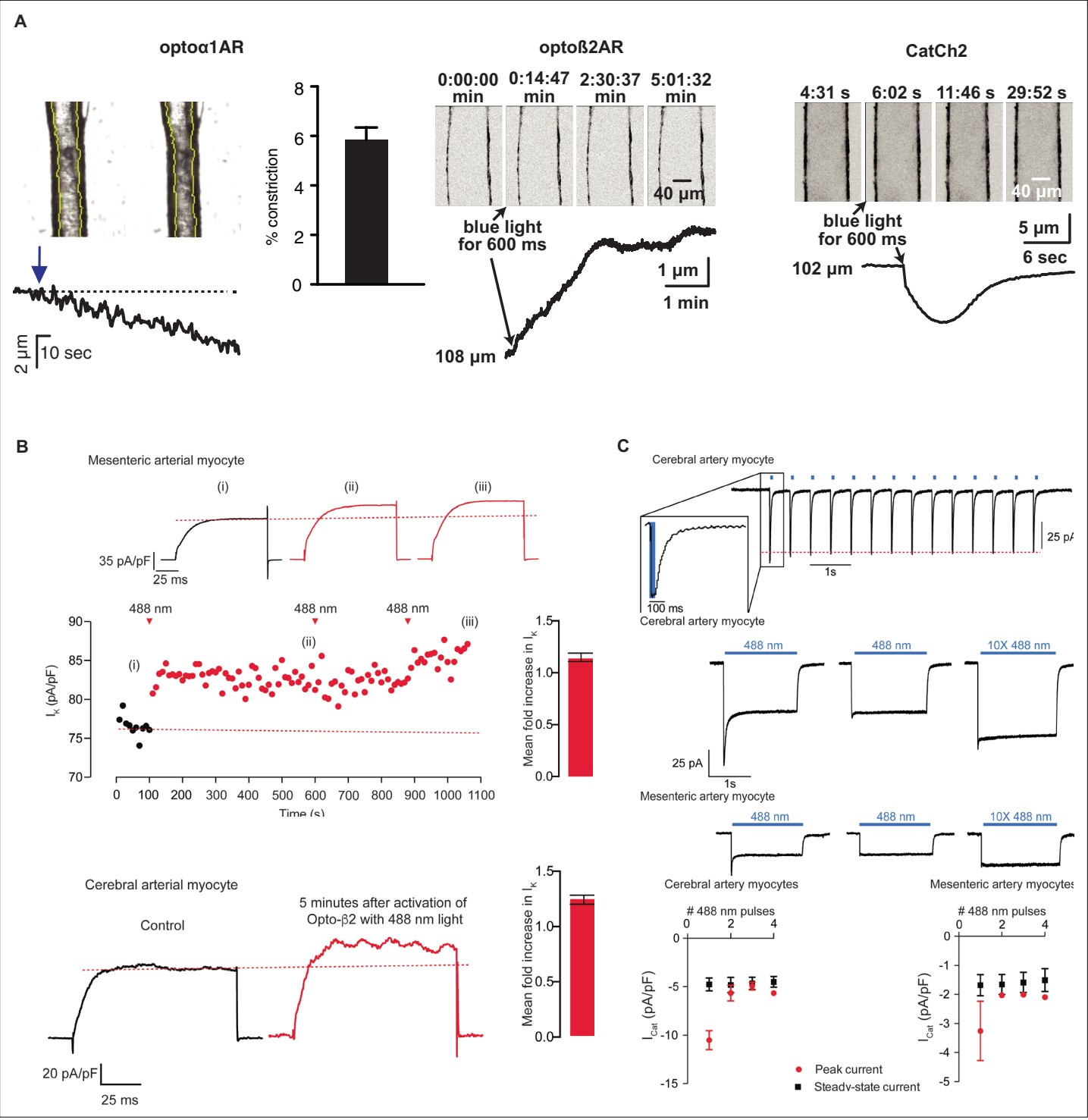

**Figure 5.** Smooth muscle optoeffector function. (**A**) Activation of optogenetic proteins in effector lines results in either vessel contraction (*Acta2*[BAC]-Optoα1AR_IRES_lacZ: left panel; *Acta2*[BAC]-CatCh2_IRES_lacZ: right panel) or relaxation (*Acta2*[BAC]-Optoß2AR_IRES_lacZ: middle panel). Source data: *Figure 5—source data 1*. (**B**) Laser activation of transgenic Optoß2AR proteins induces K currents in isolated arterial myocytes. Upper panel: mesenteric myocytes; lower panel: cerebral myocytes. Source data: *Figure 5—source data 2*. (**C**) Laser activation of transgenic CatCh2 protein induces inward membrane current. Upper panel: repeated laser pulses result in rapid inward current in cerebral myocytes. Middle panels: long pulses achieve a steady-state inward current in cerebral (upper) and mesenteric myocytes (lower) after an initial spike only on the first pulse. Bottom panel: activation of cerebral myocytes results in higher initial and steady-state current compared to mesenteric myocytes. Optoα1AR and Optoß2AR data were derived from five cells isolated from three mice, and the CatCh2 data were generated from eight cells isolated from five mice.

*Figure 5 continued on next page*

*Figure 5 continued*

The online version of this article includes the following video, source data, and figure supplement(s) for figure 5:

**Source data 1.** Arterial diameter.

**Source data 2.** K currents.

**Figure supplement 1.** Light activation of wildtype control strains.

**Figure 5—video 1.** Dilation of *Acta2*[BAC]-Optoß2AR artery stimulated with blue light.

https://elifesciences.org/articles/67858/figures#fig5video1

**Figure 5—video 2.** Vasoconstriction of *Acta2*[BAC]-CatCh2 artery stimulated with blue light.

https://elifesciences.org/articles/67858/figures#fig5video2

and third pulse of light had limited additional effects, suggesting that the initial Optoß2AR-induced potentiation of $I_K$ was near maximal. In five similar experiments in mesenteric myocytes, Optoß2AR photoactivation increased $I_K$ 1.15 ± 0.04 fold at +60 mV, and 1.23 ± 0.05 fold in cerebral arterial myocytes.

Similarly, CatCh2 optical activation was examined in cerebral and mesenteric arteries by recording membrane currents at –70 mV before and after illumination (488 nm, 50 ms pulses at 1 Hz). As shown in *Figure 5C*, optical stimulation repeatedly evoked fast inward currents that terminated as soon as the laser was turned off. In the representative cerebral myocyte shown, the amplitude of the evoked current was about 75 pA, and subsequent light pulses evoked stable $I_{Cat}$ with an amplitude of 68 ± 1 pA, suggesting that under our experimental conditions there is a relatively small level of inactivation of CatCh2 channels (*Figure 5C*, upper panel). Longer light pulses produced a larger inward current (~125 pA) that quickly (~100 ms) decayed to a lower amplitude (~35 pA) plateau, remaining relatively constant as long as the laser was turned on. This $I_{Cat}$ quickly terminated (<100 ms) after the laser was turned off. Notably, a second 488 nm light pulse to the same cell evoked an $I_{Cat}$ that lacked the high-amplitude transient component of the first light pulse but had the amplitude and kinetics of the steady-state $I_{Cat}$ in the first pulse. Similar observations were made in isolated mesenteric myocytes (*Figure 5B*, middle panels). The activation of cerebral myocytes resulted in higher currents compared to mesenteric arteries and repeated pulses resulted in dampened initial inward current (*Figure 5B*, bottom panels). Thus, *Acta2* optoeffector lines express functional chimeric receptors and channel proteins, enabling optical modulation of smooth muscle signaling in a variety of experimental conditions.

## Endothelial optogenetic mice

To enable studies of the interactions between endothelium and surrounding perivascular cells, we created three BAC transgenic optogenetic effector and one fluorescent reporter mouse lines using the endothelial cell-specific *Cdh5* gene locus. As with smooth muscle, Optoα1AR, Optoß2AR, and CatCh2 cDNAs and LacZ ORF were targeted to the *Cdh5* initiation codon, with an IRES and the LacZ sequence downstream of the optoeffectors. X-gal staining revealed wide expression in endothelial cells of the heart (*Figure 6A*), brain (*Figure 6—figure supplement 1A*), and other tissues (*Figure 6—figure supplement 1B*). Similarly, GCaMP8-selective expression in endothelial cells was achieved in *Cdh5*[BAC]-GCaMP8 mice (*Figure 6B*, *Figure 6—figure supplement 1C*). The endothelial GCaMP8 signal was sufficiently bright to be observed in coronary microvasculature using intravital two-photon microscopy (*Figure 6C*).

To confirm functional expression of optoeffector proteins, we deployed several analyses. First, we examined endothelium-induced vasodilation that occurs secondary to the release of $Ca^{2+}$ and synthesis of NO in EC (*Straub et al., 2014*). Optical activation of femoral arteries in *Cdh5*[BAC]-Optoα1AR_IRES_lacZ mice in vivo resulted in marked vasodilation, whereas laser stimulation alone caused a slight vasoconstriction in control arteries (*Figure 7A*). Similar results were obtained in cremaster arteries, but not in control arteries (*Figure 7—figure supplement 1*). We also examined the propagation of arterial dilation that occurs following muscarinic stimulation (*Tallini et al., 2007*) and is difficult to separate from agonist diffusion (*Welsh and Segal, 1998*). Laser stimulation triggered bidirectional arterial dilation at points distal to the laser beam, with a delay in the onset of vasodilation and attenuation of the amplitude (*Figure 7B*). Vasodilation was triggered at 32.2 mW, but not at lower laser powers (*Figure 7C*). Photoactivation of chimeric ß2 adrenergic receptors in Optoß2AR mice also triggered

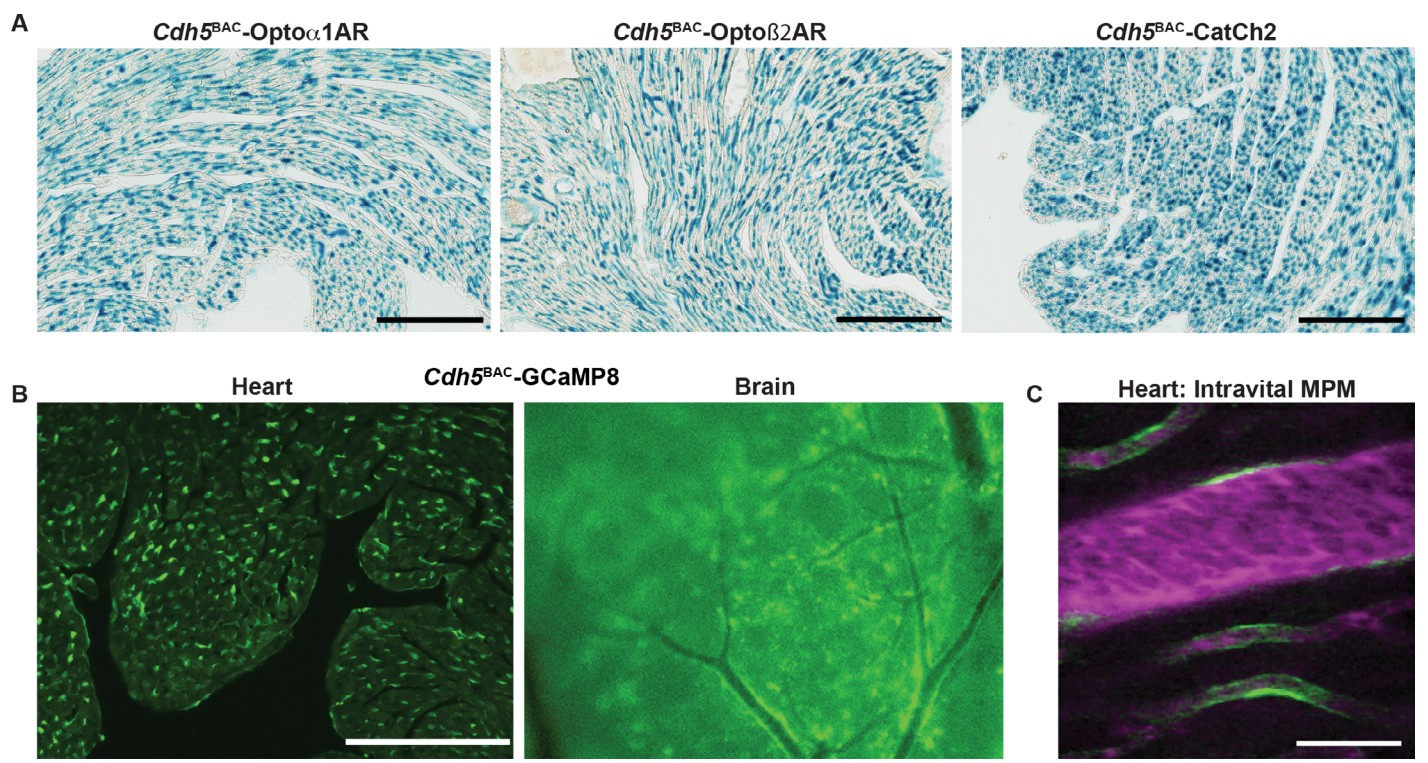

**Figure 6.** Endothelial cell optoeffector and optosensor expression. (**A**) X-gal staining of heart cryosections from *Cdh5*BAC-Optoα1AR_IRES_lacZ, *Cdh5*BAC-Optoß2AR_IRES_lacZ, and *Cdh5*BAC-CatCh2_IRES_lacZ mice. Scale bars: 200 μm. (**B**) Native fluorescence of GCaMP8 protein in heart cryosection (left panel) and brain dorsal surface (right panel). Scale bar: 200 μm. (**C**) Intravital two-photon imaging of *Cdh5* BAC-GCaMP8 mice shows endothelial GCaMP8 (green) labeling of coronary microvasculature. Intravenous Texas Red-conjugated 70 kDa dextran identifies the vessel lumen (magenta). All images shown are representative images from three animals unless otherwise specified.

The online version of this article includes the following figure supplement(s) for figure 6:

**Figure supplement 1.** *Cdh5*BAC lines.

vasodilation (*Figure 7D*). Interestingly, ß2AR-mediated dilation did not result in conducted dilation distal to the point of laser stimulation as observed in Optoα1AR mice.

## Other optogenetic lines

The gap junction protein connexin40 (*Gja5*) is expressed in specialized cells of the cardiac conduction system and in arterial endothelial cells, and expression in embryonic myocytes is critical for heart function prior to development of a mature conduction system (*Delorme et al., 1995*). To facilitate improved studies of embryonic and adult cardiac conduction defects, as well as physiological vaso-motor control, we created a *Gja5*BAC-GCaMP_GR mouse line. The ratiometric green-red fusion protein is expressed in arterial endothelia from the kidney, the ventricular conductance system and atrial myocytes (*Figure 8A*), and in arterial endothelial cells (*Figure 8—figure supplement 1A*). To extend studies to the sympathetic nervous system, we used the dopamine ß-hydroxylase locus to target opto-genetic effectors to peripheral neurons and adrenal medulla cells (reviewed in *Gonzalez-Lopez and Vrana, 2020*). X-gal staining of adult brain and adrenal glands of *Dbh*BAC-CatCh2_IRES_lacZ demon-strated the expression of LacZ in the *locus coeruleus* (*Figure 8B*) and adrenal medulla (*Figure 8—figure supplement 1B*). BAC DNA from the aldehyde dehydrogenase family 1 L1 (*Aldh1l1*) gene locus was similarly used to target glial cells. *Aldh1l1*BAC-Optoα1AR_IRES_lacZ mouse express in cerebellar (*Figure 8C*) and olfactory regions in the brain and the kidney (*Figure 8—figure supplement 1C*). The expression of Optoα1AR protein in immune cells was targeted using a BAC containing the *Lck* gene to facilitate studies on InsP$_3$ activation in T-lymphocytes. X-gal staining for LacZ expression indicates optoeffector expression in the spleen (*Figure 8D*). Mice expressing GCaMP8.1 protein in Foxj1$^+$ cells were developed using BAC RP23-294J17 (*Table 1*). Immunostaining analysis with anti-GFP antibody

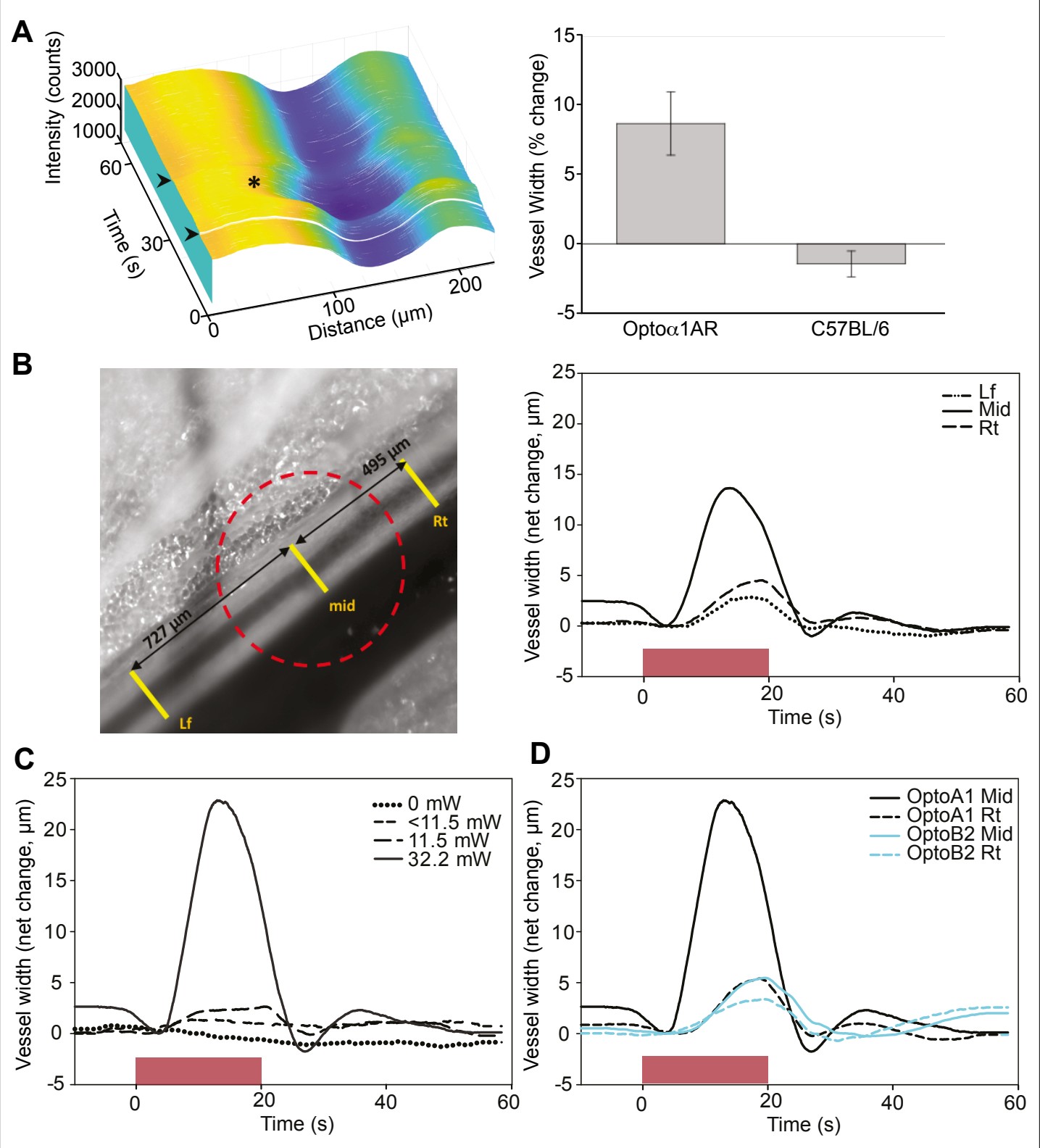

**Figure 7.** Endothelial cell optoeffector function. (**A**) Laser stimulation of femoral artery in *Cdh5*[BAC]-Optoα1AR induces vasodilation. Left panel: MATLAB-generated 3D rendering of vessel width and intensity over time. The arrowheads mark the start and end of laser activation, and the asterisk marks dilation of the artery. Right panel: comparison of peak change in vessel width between *Cdh5*[BAC]-Optoα1AR_IRES_lacZ and C57BL/6J control mice. ***p=0.003, Holm–Sidak (± SE). Optoα1AR data were derived from five femoral arteries from five animals, and the C57BL/6J data were derived from

*Figure 7 continued on next page*

*Figure 7 continued*

five femoral arteries from three animals. Source data: *Figure 7—source data 1*. (**B**) Activation of Optoα1AR protein results in migration of vasodilation away from the laser irradiation point. Left panel: brightfield image of the femoral artery with positions of the mid, right (Rt), and left (Lf) analysis positions indicated. Red dashed circle indicates laser irradiation spot on the vessel. Right panel: change in vessel width over time. Source data: *Figure 7—source data 2*. (**C**) Vessel relaxation is dependent on laser intensity. Source data: *Figure 7—source data 3*. (**D**) Comparison of vasodilation induced by Optoα1AR and Optoβ2AR proteins with laser power at 32.2 mW. Source data: *Figure 7—source data 4*. The data in (**B–D**) represent experiments with three animals.

The online version of this article includes the following source data and figure supplement(s) for figure 7:

**Source data 1.** Arterial diameter.

**Source data 2.** Arterial diameter.

**Source data 3.** Arterial diameter.

**Source data 4.** Arterial diameter.

**Figure supplement 1.** Laser activation of cremaster artery from *Cdh5*BAC-Optoα1AR leads to vessel dilation.

**Figure supplement 1—source data 1.** Arterial diameter.

indicated GCaMP8.1 expression in the bronchioles, trachea, and the testis (*Figure 8E*). We constructed the *Sftpc*BAC-GCaMP8 mouse to target reporter expression in type II alveolar cells. Anti-GFP antibody analysis of lung sections indicated wide distribution of GCaMP8 expression (*Figure 8—figure supplement 1D*).

## Discussion

We have generated 21 lines of optogenetic sensor and effector mice, in which the expression of the optogene is lineage specified in cardiovascular and related cell lineages (*Table 1*), and sensor/effector pairs can be combined through simple, monoallelic crosses. More broadly, the lines facilitate the combination of two technologies, each with enormous potential for use in the dissection of signaling processes in complex multicellular systems: genetically encoded optical sensors (*Kotlikoff, 2007*; *Nakai and Ohkura, 2003*; *Tsien, 2003*) and effectors (*Fenno et al., 2011*). Individually, these technologies have advanced our understanding of physiological in vivo processes; combined, biallelic effector/reporter mice constitute a novel platform for the activation and detection of cellular signaling in vivo. These new mouse lines comprise a publicly available molecular toolbox that enables investigators to generate physiologically relevant mono- or multiallelic transgenic mice tailored to their research interests, without the need for requisite technical expertise and financial commitment associated with constructing transgenic mice. The monoallelic nature of these lines also facilitates crosses with mouse models of cardiovascular disease. All mice are available as part of a collaborative effort between Cornell/National Heart Lung Blood Resource for Optogenetic Mouse Signaling (CHROMus) and The Jackson Laboratory.

Previous approaches have developed, or exploited, available *Cre* recombinase driver mice to confer tissue-specific expression of fluorescent markers or optogenetic sensors downstream of LoxP-STOP-LoxP sequences (*Madisen et al., 2010*; *Zariwala et al., 2012*). While this genetic strategy has the advantage of using previously characterized Cre lines and strong promoter sequences to drive optogenetic protein expression, there are several important drawbacks to this approach. These include variable penetration of the Cre-mediated recombination at LoxP sites within the target cells, limited specificity of non-BAC Cre promoters (*Madisen et al., 2010*), and the inefficient process of producing biallelic mice for experiments. The latter drawback excludes, for practical purposes, one of the most important advantages of the use of optogenetic mice – crosses with genetic strains or mutant lines that mimic human disease. The use of Cre/Lox approaches for such experiments requires complicated and inefficient breeding strategies. An additional drawback of a Cre/Lox approach relates to the requirement for constant, high transcriptional throughput in order to achieve sufficient optical sensitivity and observable fluorescent intensity. Reported optogenetic responder lines are often driven by non-specific minimal promoters and may not have sufficient transcriptional strength in all lineages sufficient for real-time optical experiments. The strains reported here provide direct, high-level tissue-specific expression of optogenetic proteins and enable simple mating strategies for the development

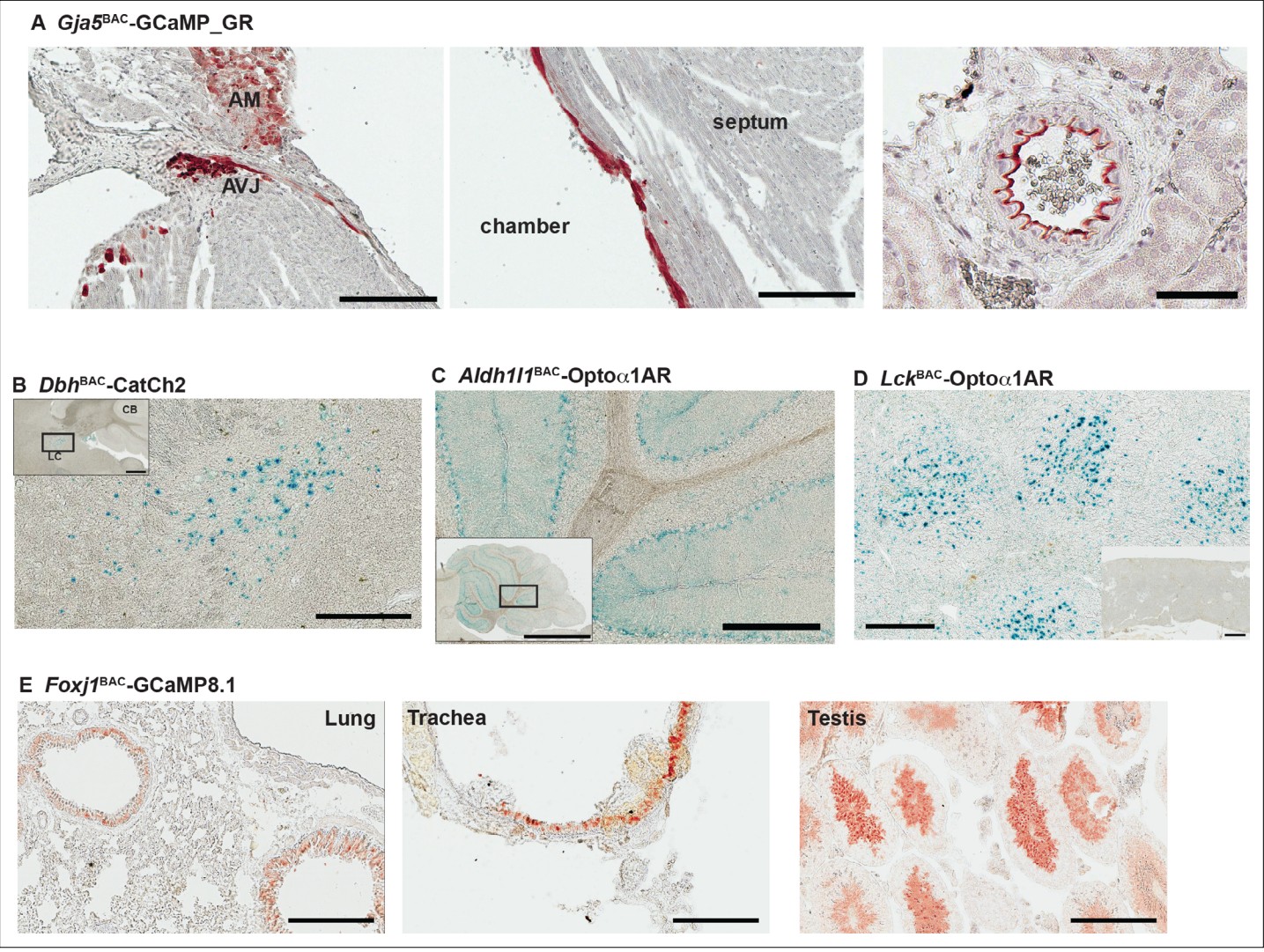

**Figure 8.** Characterization of other CHROMus mouse lines. (**A**) Anti-GFP immunohistochemistry (IHC) of *Gja5*<sup>BAC</sup>-GCaMP_GR mouse. The modified GCaMP3 protein is expressed in the atrioventricular junction (AVJ) region, atrial myocytes (AM) (left panel) and cells lining the endocardium in the adult heart (middle panel; scale bars: 200 µm) and the renal artery (right panel; scale bar: 60 µm). (**B**) X-gal staining of adult brain sagittal section from *Dbh*<sup>BAC</sup>-CatCh2_IRES_lacZ mouse showing LacZ expression in the *locus coeruleus* region. Box outlines area of enlargement. CB: cerebellum; LC: *locus coeruleus*. Scale bars; top panel: 500 µm; bottom panel: 200 µm. (**C**) X-gal staining of cerebellar region from *Aldh1l1*<sup>BAC</sup>-Optoα1AR_IRES_lacZ mouse. Box outlines area of enlargement. Scale bars; top panel: 2 mm; bottom panel: 200 µm. (**D**) X-gal staining of spleen from *Lck*<sup>BAC</sup>-Optoα1AR_IRES_lacZ mouse. Inset image shows X-gal staining from a wildtype littermate control. Scale bars: 300 µm; inset: 600 µm. (**E**) Anti-GFP IHC of lung, trachea, and testis tissues from *Foxj1*<sup>BAC</sup>-GCaMP8.1 mouse. Scale bars: 200 µm. All images shown are representative images from three animals unless otherwise specified.

The online version of this article includes the following figure supplement(s) for figure 8:

**Figure supplement 1.** Transgene expression in other CHROMus lines.

of biallelic mice. Each line has been analyzed for expression and lines with off-target or low levels of expression have been eliminated, and each line has been bred for multiple generations.

On the other hand, the lines reported here result from the random integration of large DNA elements into the mouse genome in a stable manner, whereas many (but not all) Cre-driver and LoxP flanked responder lines result from targeted integration within a known locus. Thus a drawback of CHROMus lines is the unknown site of integration, which could result in integration site-dependent effects on the transcription of other genes, or impact the transcriptional fidelity of the inserted optogenetic cDNA due to positional effects. While we have verified the insertion of BAC or cDNA sequences in these lines, and confirmed the viable breeding of the reported lines, these effects cannot be completely

known and thus the lines should be used where they provide functional value, but may not be useful in all circumstances. One example is the *Sftpc*^BAC^-GCaMP8 line, which expresses the GCaMP8 protein in type II alveolar cells, but a subset of type II cells did not express the reporter protein. Also, we noted a lack of X-gal staining in the central nervous system of *Hcn4*^BAC^-CatCh mice, but this expression pattern has been previously reported for HCN4 mRNA and protein (*Kouranova et al., 2008*). Finally, we note that laser stimulation of the left ventricle in the *Hcn4*^BAC^-CatCh mouse failed to induce pacing.

Separate targeting of cardiac conducting cells and myocytes enabled light-activated heart activation, and we demonstrate activation of the heart either through stimulation of cells within the SA node (*Figure 2*) or extra-conduction system activation equivalent to premature ventricular contractions (*Figure 3*). By crossing nodal optoeffector (*Hcn4*^BAC^-CatCh2_IRES_LacZ) and myocyte sensor (*Myh6*-GCaMP8 strains) mice, we demonstrate the feasibility of combinatorial optogenetics and report the ability to activate and detect heart signaling (*Figure 2H*, *Figure 2—video 2*). We also demonstrate the ability to selectively examine cardiac conduction using *Hcn4*^BAC^-GCaMP8 mice and their robust signaling of conduction events in embryonic, neonatal, and adult hearts (*Figure 2*, *Figure 2—figure supplement 2*). The cardiac GCaMP8 sensor lines represent several improvements as they exploit recent structure-based improvements in circular permutation of GFP (*Ohkura et al., 2012b*; *Tian et al., 2009*), resulting in higher brightness and dynamic range. Conduction failures result in the implantation of more than 200,000 pacemakers annually in the United States alone (*Mond and Proclemer, 2011*), and the ability to selectively initiate cardiac conduction at local sites within the heart (optoeffector lines), either through activation of specialized conducting cells or myocytes, and to monitor activation of the conduction system (optosensor lines), confers significant experimental advantages both in understanding conduction system failure and strategies to develop biological pacemakers (*Meyers et al., 2016*; *Nyns et al., 2019*; *Vedantham, 2015*). Similar approaches can be envisioned for studies of heart failure, a major cause of mortality worldwide (*Bui et al., 2011*). Two areas of specific value would be the understanding of cardiomyopathy-induced arrhythmias (*Asimaki et al., 2014*) and cell and gene-based approaches to therapy (*Nakamura and Murry, 2019*).

The development of mouse lines with robust expression of optical effectors and sensors in smooth muscle and endothelial cells provides similar experimental advantages for vascular biology. We report the first mice conferring lineage-specific expression of sensors that allow light-activated depolarization through CatCh2 channels or release of intracellular $Ca^{2+}$ through Optoα1AR, chimeric α-adrenoceptors, as well as relaxation of smooth muscle through photoactivation of Optoß2AR, chimeric ß2 receptors. Interestingly, we observed rhythmic endothelial cell signaling during the cardiac cycle, enabling dissection of mechanisms that regulate blood flow, matching perfusion to metabolic demand (*Duncker and Bache, 2008*). Similar approaches have led to important advances in the understanding of metabolic vascular coupling in the brain and other arterioles (*Gonzales et al., 2020*; *Sonkusare et al., 2012*; *Sonkusare et al., 2014*), but studies in the heart face additional imaging challenges associated with imaging the beating heart. We used stabilized optical windows and deconvolution approaches to address these challenges (*Figures 2 and 3*). The creation of lines that enable ratiometric measurements (*Acta2*^BAC^-GCaMP_GR and *Acta2*^BAC^-GCaMP8.1_Vermilion; *Figure 4*) adds additional experimental advantages in this regard. Similarly, the ability to initiate synthesis of critical smooth muscle second messengers through the expression of light-sensitive chimeric receptor proteins fused to catalytic domains of α1 and ß2 adrenoceptors (*Airan et al., 2009*) provides important capabilities to probe vascular control mechanisms in smooth muscle tissues, and the ability to control the synthesis in an individual cell type.

We report additional lines that enable studies of sympathetic neurons, glial cells, and type II alveolar cells. These lines show robust, tissue-specific expression of the optoeffector or optosensor proteins, enabling similar studies of neurovascular and lung cell signaling and provide important advantages for the study of neurosecretion, neuromodulation, and excitation/secretion coupling.

Finally, we anticipate that CHROMus mice, either as effector/detector pairs or individually, will have great utility when crossed with existing mouse models of human disease. Such models exist for many cardiovascular, neurovascular, airway, and immune diseases, and the mouse lines reported here provide a direct way to apply powerful optogenetic techniques to the study of pathophysiology and the assessment of treatment.

In summary, BAC transgenic optogenetic effector and sensor lines provide novel experimental platforms for real-time in vivo investigations of molecular signaling in complex physiological systems.

The mice are available to experimenters through JAX and represent a tool kit that can advance the use of optogenetics in a wide array of experimental investigations.

# Materials and methods

## Key resources table

| Reagent type (species) or resource | Designation | Source or reference | Identifiers | Additional information |
|---|---|---|---|---|
| Antibody | Rabbit polyclonal anti-GFP | Santa Cruz | RRID:AB_641123 | IHC (1:500) |
| | CHROMus ID and strain name | Jackson # | | Jackson Laboratory strain name |
| Genetic reagent (*Mus musculus*) | B3-*Acta2*<sup>BAC</sup>-RCaMP1.07 | 28345 | MGI:5750184 | Tg(RP23-370F21-RCaMP1.07)B3-3Mik/J |
| Genetic reagent (*M. musculus*) | B4-*Myh6*-GCaMP8 | 33341 | RRID:IMSR_CHROMUS:B4 | Tg(*Myh6*-GCaMP8)B4-10Mik/J |
| Genetic reagent (*M. musculus*) | B5-*Myh6*-CatCh2_IRES_lacZ | 30334 | RRID:IMSR_CHROMUS:B5 | Tg(*Myh6*-COP4*L132C,-lacZ)B5-7Mik/J |
| Genetic reagent (*M. musculus*) | B6-*Acta2*<sup>BAC</sup>-Optoa1AR_IRES_lacZ | 28346 | RRID:IMSR_CHROMUS:B6 | Tg(*Acta2*-RHO/ADRA1,-lacZ)B6-1Mik/J |
| Genetic reagent (*M. musculus*) | B7-*Acta2*<sup>BAC</sup>-Optoß2AR_IRES_lacZ | 28347 | RRID:IMSR_CHROMUS:B7 | Tg(*Acta2*-RHO/Adrb2,-lacZ)B7-5Mik/J |
| Genetic reagent (*M. musculus*) | B8-*Acta2*<sup>BAC</sup>-CatCh2_IRES_lacZ | 28348 | RRID:IMSR_CHROMUS:B8 | Tg(*Acta2*-COP4*L132C,-lacZ)B8-3Mik/J |
| Genetic reagent (*M. musculus*) | B10-*Hcn4*<sup>BAC</sup>-GCaMP8 | 28344 | MGI:5750171 | Tg(*Hcn4*-GCaMP8)B10-3Mik/J |
| Genetic reagent (*M. musculus*) | B14-*Acta2*<sup>BAC</sup>-GCaMP_GR | 25406 | MGI:5693405 | Tg(RP23-370F21-GCaMP3*/mCherry)1Mik/J |
| Genetic reagent (*M. musculus*) | B15-*Gja5*<sup>BAC</sup>-GCaMP_GR | 30333 | MGI:6281638 | Tg(*Gja5*-GCaMP3*/mCherry)B15-2Mik/J |
| Genetic reagent (*M. musculus*) | B16-*Dbh*<sup>BAC</sup>-CatCh2_IRES_lacZ | 32666 | MGI:6361520 | Tg(*Dbh*-COP4*L132C,-lacZ)B16-4Mik/J |
| Genetic reagent (*M. musculus*) | B17-*Hcn4*<sup>BAC</sup>-CatCh2_IRES_lacZ | 33344 | RRID:IMSR_CHROMUS:B17 | Tg(*Hcn4*-COP4*L132C,-lacZ)B17-2Mik/J |
| Genetic reagent (*M. musculus*) | B19-*Acta2*<sup>BAC</sup>-GCaMP2 | 25405 | MGI:5693385 | Tg(RP23-370F21-GCaMP2)6Mik/J |
| Genetic reagent (*M. musculus*) | B20-*Cdh5*<sup>BAC</sup>-GCaMP8 | 33342 | RRID:IMSR_CHROMUS:B20 | Tg(*Cdh5*-GCaMP8)B20-6Mik/J |
| Genetic reagent (*M. musculus*) | B22-*Lck*<sup>BAC</sup>-Optoa1AR_IRES_lacZ | 33705 | MGI:6361523 | Tg(*Lck*-RHO/ADRA1,-lacZ)B22-1Mik/J |
| Genetic reagent (*M. musculus*) | B23-*Sftpc*<sup>BAC</sup>-GCaMP8 | 32885 | RRID:IMSR_CHROMUS:B23 | Tg(*Sftpc*-GCaMP8)B23-5Mik/J |
| Genetic reagent (*M. musculus*) | B26-*Cdh5*<sup>BAC</sup>-Optoa1AR_IRES_lacZ | 33343 | MGI:6294081 | Tg(*Cdh5*-RHO/ADRA1,-lacZ)B26-1Mik/J |
| Genetic reagent (*M. musculus*) | B27-*Cdh5*<sup>BAC</sup>-Optoß 2AR_IRES_lacZ | 32889 | MGI:6324937 | Tg(*Cdh5*-RHO/Adrb2,-lacZ)B27-3Mik/J |
| Genetic reagent (*M. musculus*) | B28-*Cdh5*<sup>BAC</sup>-CatCh2_IRES_lacZ | 33345 | MGI:6324934 | Tg(*Cdh5*-COP4*L132C,-lacZ)B28-2Mik/J |
| Genetic reagent (*M. musculus*) | B34-*Acta2*<sup>BAC</sup>-GCaMP8.1_mVermilion | 32887 | MGI:6324943 | Tg(*Acta2*-GCaMP8.1/mVermilion)B34-4Mik/J |
| Genetic reagent (*M. musculus*) | B35-*Foxj1*<sup>BAC</sup>-GCaMP8.1 | 32888 | MGI:6324940 | Tg(*Foxj1*-GCaMP8.1)B35-3Mik/J |
| Genetic reagent (*M. musculus*) | B36-*Aldh1l1*<sup>BAC</sup>-Optoa1AR_IRES_lacZ | 33706 | MGI:6361527 | Tg(*Aldh1l1*-RHO/ADRA1,-lacZ)B36-3Mik/J |

Information on all mouse lines and the CHROMus project can be found at https://chromus.vet.cornell.edu. CHROMus mice are available from the Jackson Laboratory or IMSR.

## Construction of transgenic mice

Mouse lines were created by injection of either homologously recombined bacterial artificial chromosomes (BACs) (*Tallini et al., 2006b*), or previously established promoter constructs with desired effector or reporter, into fertilized single-cell eggs. All constructs are listed in *Table 1*. The optogenetic effector lines contained either optoα1AR (*Airan et al., 2009*), optoß2AR (*Airan et al., 2009*), or CatCh2 (*Zhang et al., 2006*) genes linked to downstream IRES-lacZ for simple identification of transgene expression. The reporter lines contained either EGFP- or RFP-derived GECIs as a single fluorophore (GCaMP8 [*Ohkura et al., 2012b*], or GCaMP8.1 [GCaMP8 with M144I mutation], or RCaMP1.07 [*Ohkura et al., 2012a*]) or as a fusion protein of green and red fluorophores GCaMP_GR (*Shui et al., 2014*) or GCaMP8.1_mVermilion. BAC transgenes were constructed by homologous recombination to position the desired transgene in frame with the initiation codon of the native gene as previously described (*Tallini et al., 2006b*). Purified BAC transgene plasmids were microinjected into single-cell embryos at the Cornell University Transgenic Core Facility or the UC Irvine Transgenic Mouse Facility. Pups were screened for the desired transgene by PCR analysis using primers listed in online *Table 1*, backcrossed to C57BL/6J and further screened by immunohistochemistry, X-gal staining, or fluorescence imaging using standard methods. Briefly, tissues were either fixed overnight in 4% paraformaldehyde (PFA) at 2–8°C, rinsed in 1× phosphate-buffered saline (PBS) and equilibrated in 30% sucrose/1× PBS at 2–8°C (for immunohistochemistry), or fixed for 2 hr in 4% PFA on ice (for X-gal staining, see Supplemental materials) and processed to 30% sucrose. Anti-GFP antibody was used as previously described (*Jesty et al., 2012*). For fluorescence imaging, sections were washed three times in 1× PBS, mounted with VectaShield with DAPI (Vector Laboratories), and imaged using a Leica DMI6000B inverted microscope. Monochrome images were colorized using ImageJ software.

## X-gal staining

LacZ expression from the optogenetic transgene constructs were screened by X-gal staining (1 mg/mL in 1× PBS; 1 mM $MgCl_2$; 250 µM potassium ferricyanide; 250 µM potassium ferrocyanide) at 37°C for either 4 hr or overnight. For whole-embryo staining, the embryos were fixed in 4% PFA (pH 7.0) for 30 min on ice and washed three times in detergent wash buffer (0.1 M phosphate buffer, pH 7.3; 2 mM $MgCl_2$; 0.01% Na deoxycholate; 0.02% NP-40) for 20 min. Whole tissues were fixed in 4% PFA for 2 hr on ice, washed three times in 1× PBS for 10 min and stained in X-gal. For frozen sections, the fixed tissues were dehydrated in 30% sucrose (1× PBS), embedded in OCT, and sectioned at 8 µm. The sections were rinsed in 1× PBS for 5 min (3×) and stained in X-gal.

## Whole heart clearing

The neonatal heart was cleared using BABB as described (*Joyner and Wall, 2008*) and imaged using Olympus OV100 digital camera mounted to a Leica stereoscope.

## Ex vivo arterial imaging

Freshly isolated mid cerebral or mesenteric arteries were cannulated on glass pipettes mounted in a 5 mL myograph chamber and pressurized as described previously (*Prada et al., 2019*), and only arteries that constricted more than 30% to isosmotic physiological salt solution containing 120 mM KCl were used. Optogenetic effector proteins were activated at 473 nm using an Olympus FV1000 microscope with a SIM scanner, which enables photostimulation. To avoid unintended activation of genetic proteins by ambient light, the isolated vessels were kept in the dark or under red light. Arterial diameter was determined from stacks of tiff images using the Myotracker video-edge detection plugin in ImageJ (*Fernández et al., 2014*).

## Ex vivo heart imaging

Mice were given 100 U of heparin IV prior to euthanasia by pentobarbital (120 mg/kg, IP). The heart was quickly excised with aorta and vena cava vessels attached and rinsed in ice-cold Tyrode's buffer (137 mM NaCl; 2.7 mM KCl; 10 mM HEPES; 1.4 mM $CaCl_2$; 1.0 mM $MgCl_2$; 0.4 mM $NaH_3PO_4$; 12 mM $NaHCO_3$; 5.5 mM D-glucose; pH 7.4). The rinsed heart was cannulated and perfused with warm

Tyrode's buffer (37°C , equilibrated with 95% $O_2$/5% $CO_2$). GCaMP8 fluorescence was captured using an Andor-iXon CCD camera mounted on an Olympus MVX10 fluorescent microscope (ex: 472/30 nm; em: 520/35 nm).

### In vivo heart stimulation

Mice were anesthetized with ketamine (100 mg/kg) and xylazine (10 mg/kg) via intraperitoneal injection, intubated with a 22-gauge cannula, ventilated (95 breath/min, 12 cm $H_2O$ end-inspiratory pressure; CWE SAR-830/P ventilator), and maintained on 1.5% isoflurane. 5% glucose (0.1 mL/10 g) and atropine sulfate (5 µg/100 g body weight) were injected subcutaneously and body temperature was maintained at 37.5°C with a heating pad. The heart was surgically exposed by dissection of the sternum and diaphragm, and ribs retracted for optical access to the thoracic cavity. For optical stimulation of the SA node, mice were positioned on their back on a stereotaxic stage, the pericardial sac was removed, and the thymus gland repositioned from covering the right atrium. Illumination of the node was performed using a 105 µm core fiber (Thorlabs) coupled to a 473 nm continuous wave, diode-pumped solid-state laser system (Opto Engine LLC; BL-473-00200-CWM-SD-03-LED-F). The tip of the laser fiber was inserted through 500 µm metal tubing and clamped within a four-way axis micromanipulator that allowed positioning of the fiber tip on the anterior junction of the right atrium and superior vena cava. For stimulation of the left ventricle free wall, mice were positioned on their right side and a left thoracotomy was performed between ribs 7 and 8 to gain access to the heart. The tip of the optical fiber was coupled to a collimating lens mount that was positioned to direct the beam at the left ventricle. A customized titanium probe affixed to the stereotaxic stage was adhered (Vetbond) to the left ventricle free wall to minimize movement associated with respiratory and cardiac contractions. Electrocardiogram (ECG) electrodes were attached to 21-gauge needles inserted subcutaneously through the front and contralateral hind limb and recorded with an isolated differential amplifier (World Precision Instruments; #ISO-80). ECG and lung pressure traces were continuously monitored through an oscilloscope. Laser pulses were controlled through a TTL signal delivered by a function generator (Agilent 33210A) and signals digitized and recorded on a computer using MATLAB.

### SA node stimulation

For in vivo stimulation of the SA nodal tissue, the mouse was anesthetized with 1.5% isoflurane and positively ventilated by intubation as described above. The ECG electrodes were attached and the heart was visualized by opening the thoracic cavity. The metal tube containing the fiber optic cable was placed adjacent to the SA node. The laser was controlled as described above and the ECG recorded. For ex vivo stimulation, the fiber optic cable was positioned over the SA nodal tissue of the excised heart and laser controlled as described above.

### In vivo two-photon imaging

A left thoracotomy was performed as above and a 3D-printed titanium stabilization probe with imaging window (*Allan-Rahill et al., 2020*; *Jones et al., 2018*) attached to the left ventricle free-wall using tissue adhesive (Vetbond). Texas Red-conjugated 70 kDa dextran (3% in saline; Thermo Fisher Scientific #D1830) was injected retro-orbitally to identify the vasculature. A Ti:Sapphire laser (Chameleon, Coherent) with wavelength centered at 950 nm was used to excite indicator molecules and images collected using a custom multiphoton microscope equipped with four detection channels and high-speed resonant scanners running ScanImage (*Pologruto et al., 2003*). Emission fluorescence was detected using long-pass dichroic mirrors and bandpass filters for GCaMP8 (517/65 nm) and Texas Red (629/56). Water was placed within a rubber O-ring of the stabilization probe to allow immersion of the microscope objective (Olympus XLPlan N 25 × 1.05 NA). ECG and respiratory signals were collected while imaging z-stacks (50–100 frames; 2 µm per z-step; 30 frames/s) in 4–5 different regions in each mouse. Image reconstruction was performed as previously described (*Jones et al., 2018*) by indexing image lines based on their acquired position within the cardiac or respiratory cycles.

### In vivo arterial imaging

Mice were anesthetized with 5% isoflurane, maintained with 2% isoflurane using a nose cone, and placed on a heated stage. The cremaster or femoral vessels were visualized and superfused with warm PSS equilibrated with 95% $O_2$-5% $CO_2$ (*Bagher and Segal, 2011*). Endothelial cells of either

cremaster or femoral vessels were stimulated with a 473 nm, continuous wave, diode-pumped solid-state laser system (Opto Engine LLC; BL-473-00200-CWM-SD-03-LED-F) at power and pulse length as specified. Optical fiber coupled to a collimating lens was attached to a three-axis micromanipulator and positioned to aim the light at the desired vessel. To quantify arterial diameter changes, vessels were illuminated with white light. Raw images from the Andor-iXon camera were converted to .tif files using ImageJ. Motion artifacts between frames were removed using MATLAB's Image Processing Toolbox prior to vessel analysis. Each frame was translated to maximize the two-dimensional correlation with respect to the first frame using the function *imregcorr*. This correlation was calculated in an area of the images not affected by light leakage from the stimulation laser in order to avoid stabilizing on the beam rather than the tissue structure. Once the image was stabilized, vessel width and sidewall intensity were measured as a function of time across each frame. MATLAB was used to interpolate the intensity profile along a user-specified line across a vessel using the *interp2* function. To improve the resulting profile, intensity was averaged over 28.05 µm. The two peaks in the profile were recorded as the sidewall intensities and the distance between the peaks recorded as the vessel width. If the profile still included significant noise, a quadratic fit was performed around each peak to find a more accurate peak location and intensity.

## ECG analysis

ECG and laser pulse recordings were loaded into MATLAB. Appropriate preprocessing was performed to (1) (when necessary) resample non-continuous recordings, (2) lowpass filter ECGs with a passband frequency of 100 Hz, (3) invert, scale, and shift ECGs to have a median value of zero and a maximum amplitude of 1, and (4) binarize laser pulse recordings. ECG R-wave peaks were detected automatically using MATLAB's *findpeaks* function, and P-waves were detected by finding the local maximum within an ~100 ms window before the R-wave. All peaks were manually curated and adjusted as necessary. Beats were classified as unstimulated if they occurred before the onset of stimulation pulses. For *Myh6*-CatCh2 recordings, beats were classified as stimulated if an R-wave occurred 10–38 ms after a laser pulse rising edge, and for *Hcn4*[BAC]-CatCh2 recordings, beats were classified as stimulated if a P-wave occurred 38–52 ms after a laser pulse falling edge. Beats occurring after onset of stimulation but that did not meet the timing criteria were excluded from analysis. R-wave duration was calculated as the time interval between zero crossings on either side of the R-wave. PR interval was calculated as the time interval between peak P-wave and the first zero crossing of the QRS complex. PR intervals and R-wave durations were imported into Prism to generate violin plots and calculate p-values using the Mann–Whitney test.

## ImageJ

The monochrome fluorescent digital images obtained from Leica DMI6000B/Q-imaging or Olympus MVX-10/Andor-iXon CCD cameras were colorized using the Lookup Table command in ImageJ.

## MATLAB data transformation and statistical analysis

The raw data for vessel width from MATLAB analysis were transformed using the running average algorithm in SigmaPlot 11 and the transformed data were copied to Excel spreadsheet. The net change in vessel width was calculated by finding the minimum width in the first 10 s after laser activation using the MIN function in Microsoft Excel and subtracting this value from SigmaPlot transformed data.

The % change in vessel width was calculated by first determining the maximum width during laser activation using the MAX function in Microsoft Excel and using the formula ((MAX − MIN)/MIN) * 100. The % change in vessel width from five femoral vessels from *Cdh5*[BAC]-Optoα1AR and C57BL/6J animals were determined and compared using the Holm–Sedak method (SigmaPlot v 11).

## Patch-clamp electrophysiology

All current recordings were acquired at room temperature using an Axopatch200B amplifier and a Digidata1440 digitizer (Molecular Devices, Sunnyvale, CA). Borosilicate patch pipettes were pulled and polished to resistance of 3–6 MΩ for all experiments using a micropipette puller (model P-97, Sutter Instruments, Novato, CA). Freshly dissociated arterial myocytes were kept in ice-cold $Mg^{2+}$-PSS before being recorded.

Voltage-gated K⁺ ($I_K$) and light-activated CatCh2 ($I_{Cat}$) currents were recorded using conventional whole-cell configuration of the voltage-clamp technique. Currents were recorded at a frequency of 50 kHz and low-pass filtered at 2 kHz. $I_K$ and $I_{Cat}$ were recorded while cells were exposed to an external solution composed of 130 mM NaCl, 5 mM KCl, 3 mM $MgCl_2$, 10 mM glucose, and 10 mM HEPES adjusted to 7.4 using NaOH. The pipette solution was composed of 87 mM K-aspartate, 20 mM KCl, 1 mM $CaCl_2$, 1 mM $MgCl_2$, 5 mM MgATP, 10 mM EGTA, and 10 mM HEPES adjusted to 7.2 by KOH. A liquid junction potential of 12.7 mV was corrected offline. $I_K$ were activated by a series of 500 ms test pulses increasing from –70 mV to +60 mV. $I_{Cat}$ was recorded while cells were held at –70 mV before and after exposure to 488 nm light.

## Acknowledgements

We thank the Cornell Stem Cell and Transgenics Core, and the University of California, Irvine Transgenic Mouse Facility (TMF) for production of BAC transgenic mice. The UCI TMF is a shared resource funded in part by the Chao Family Comprehensive Cancer Center Support Grant (P30CA062203) from the National Cancer Institute.

## Additional information

### Funding

| Funder | Grant reference number | Author |
|---|---|---|
| National Institutes of Health | R24 HL120847 | Michael I Kotlikoff |
| American Heart Association | 13SDG17330004 | Nozomi Nishimura |
| National Institutes of Health | P01 AI102851 | Nozomi Nishimura |
| National Institutes of Health | R21 AG058173 | Nozomi Nishimura |
| National Institutes of Health | R01 HL122827 | Megan A Rizzo |
| National Institutes of Health | R01 HL121059 | Manuel F Navedo |
| National Institutes of Health | R01 HL149127 | Manuel F Navedo |
| National Institutes of Health | R01 HL152681 | L Fernando Santana |
| National Institutes of Health | OT2 OD026580 | L Fernando Santana |
| American Heart Association | 17POST3368012 | David M Small |

The funders had no role in study design, data collection and interpretation, or the decision to submit the work for publication.

### Author contributions

Frank K Lee, Data curation, Formal analysis, Investigation, Methodology, Project administration, Validation, Visualization, Writing – original draft, Writing – review and editing; Jane C Lee, Conceptualization, Data curation, Funding acquisition, Investigation, Methodology, Project administration, Resources, Supervision, Validation, Writing – review and editing; Bo Shui, Methodology, Resources; Shaun Reining, Resources; Megan Jibilian, Investigation; David M Small, Data curation, Formal analysis, Funding acquisition, Investigation, Methodology, Validation, Visualization, Writing – original draft, Writing – review and editing; Jason S Jones, Data curation, Formal analysis, Investigation, Methodology, Validation, Visualization, Writing – original draft; Nathaniel H Allan-Rahill, Formal analysis,

Visualization, Writing – review and editing; Michael RE Lamont, Formal analysis, Methodology, Visualization, Writing – original draft, Writing – review and editing; Megan A Rizzo, Funding acquisition, Resources, Writing – review and editing; Sendoa Tajada, Data curation, Investigation, Methodology, Validation, Visualization, Writing – review and editing; Manuel F Navedo, Data curation, Formal analysis, Funding acquisition, Investigation, Methodology, Validation, Visualization, Writing – review and editing; Luis Fernando Santana, Conceptualization, Data curation, Formal analysis, Funding acquisition, Investigation, Methodology, Supervision, Validation, Visualization, Writing – original draft, Writing – review and editing; Nozomi Nishimura, Conceptualization, Funding acquisition, Supervision, Visualization, Writing – original draft, Writing – review and editing; Michael I Kotlikoff, Conceptualization, Data curation, Funding acquisition, Methodology, Project administration, Supervision, Validation, Visualization, Writing – original draft, Writing – review and editing

## Author ORCIDs
Nathaniel H Allan-Rahill http://orcid.org/0000-0002-8090-8339
Megan A Rizzo http://orcid.org/0000-0001-6528-7768
Sendoa Tajada http://orcid.org/0000-0002-5138-9808
Manuel F Navedo http://orcid.org/0000-0001-6864-6594
Luis Fernando Santana http://orcid.org/0000-0002-4297-8029
Nozomi Nishimura http://orcid.org/0000-0003-4342-9416
Michael I Kotlikoff http://orcid.org/0000-0001-9879-1899

## Ethics

All animal protocols were reviewed and approved by the Cornell University Institutional Animal Care and Use Committee (IACUC) (2005-0167 to MIK, 2015-0029 to N.N) and by University of California, Davis IACUC (20738 to F.L.S) in accordance with the National Institutes of Health Guide for the Care and Use of Laboratory Animals.

## Decision letter and Author response

Decision letter https://doi.org/10.7554/eLife.67858.sa1
Author response https://doi.org/10.7554/eLife.67858.sa2

## Additional files

### Supplementary files
• Transparent reporting form

### Data availability

The mouse strains have been deposited with Jackson Laboratory. The source data files for Figures 2, 3, 5, 7 and Figure 7 - figure supplement have been provided.

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
