## [Decision Letter]

**Acceptance summary:**

The manuscript by Lee and co-workers describes the development of 21 unique transgenic mouse lines that express optogenetic sensors and effectors in a cell lineage-specific fashion. The knock-in approach allows the sensors and effectors to be rapidly combined or moved to different backgrounds, such as genetic disease models. Such manipulations are often impractical when using a Cre-based system. This constitutes a vital advantage for many studies. The new mice described here will be very powerful tools to study physiology and alteration in disease models.

**Decision letter after peer review:**

Thank you for submitting your article "Genetically Engineered Strains for Combinatorial Cardiovascular Optobiology" for consideration by *eLife*. Your article has been reviewed by 3 peer reviewers, and the evaluation has been overseen by a Reviewing Editor and Didier Stainier as the Senior Editor. The following individual involved in review of your submission has agreed to reveal their identity: Scott Earley (Reviewer #1).

Essential revisions:

1) The manuscript would benefit from a transparent discussion of potential problems/limitations associated with the mice described here. For example, is there any evidence for off-target or incomplete expression of sensors or effectors? Are there any issues with the breeding of any of these lines?

2) In at least one case, the description on the Jackson Laboratory website did not exactly match the description in the Major Resource Table. Jackson stock number 025406 links to a GCaMP3- mCherry mouse, but this is identified as GCaMP5-mCherry in the Table. Please carefully verify this vital information to avoid confusion.

3) As provided for experiments using endothelial optogenetic mice, controls using similar laser direction and parameters should be included for each experimental test of the optogenetic activators.

4) A major advantage of using mice of the types described is the ease in which disease models, infection, additional genetic manipulations, environmental factors, aging, etc, can be superimposed upon the mice carrying the optogenetic activators and sensors. This point is mentioned only briefly in the discussion. The manuscript would be strengthened if an example of such an application could be provided in the manuscript. It would be nice to see a graded effect of optogenetic stimulation to provide more evidence of the utility of the approach.

5) The probes and effectors are constitutively expressed throughout development. This is different from an inducible LoxP-Stop-LoxP approach. While clearly all the animals are viable, some discussion would be helpful that addresses how expression of a probe (for example a Ca indicator) might potentially alter function. Have the Authors assessed that this is not an issue experimentally?

*Reviewer #1 (Recommendations for the authors):*

1. The manuscript would benefit from a transparent discussion of potential problems/limitations associated with the mice described here. For example, is there any evidence for off-target or incomplete expression of sensors or effectors? Are there any issues with the breeding of any of these lines?

Anecdotally, the Acta2-GCaMP5-mCherry mice failed to breed in my laboratory, and I've heard from at least one other investigator who had the same issue. I have discussed this with some of the authors of the current study. Have any of the authors of the study encountered such problems with any of the other lines? It would be helpful to future users to disclose these issues and provide practical guidance.

2. In at least one case, the description on the Jackson Laboratory website did not exactly match the description in the Major Resource Table. Jackson stock number 025406 links to a GCaMP3- mCherry mouse, but this is identified as GCaMP5-mCherry in the Table. I didn't check them all. I suggest that the authors carefully verify this vital information to avoid confusion. Also, from the Jackson laboratory website, most, if not all, of these strains are cryopreserved. Recovery takes several months and is expensive. It might be helpful to potential users to disclose this limitation.

*Reviewer #2 (Recommendations for the authors):*

The paper is well-written and thought provoking. The study is essentially a methods papers with some, if not all, of the mice described having been used in previous investigation-based studies.

As provided for experiments using endothelial optogenetic mice, controls using similar laser direction and parameters should be given for each experimental test of the optogenetic activators.

While the breeding tactics to generate experimentally useful chimeras are clear, it would also be helpful to compare relative expression/signal strengths for Cre recombinase driver mice vs. the monoallelic mice promoted in the present study. Comments about breeding success/difficulties would also be useful for those planning experiments using combinations of the mice from CHROMus.

A major advantage of using mice of the types described is the ease in which disease models, infection, additional genetic manipulations, environmental factors, aging, etc, can be superimposed upon the mice carrying the optogenetic activators and sensors. This point is mentioned only briefly in the discussion, but frankly it is of such importance for future investigation that an example of such an application should be provided in the manuscript.

*Reviewer #3 (Recommendations for the authors):*

I have a few suggestions:

While the potential for this approach is vast and the data presented validates the approach, no new science or optogenetic/probes are really presented in this paper.

It would be nice to see a graded effect of optogenetic stimulation to provide more evidence of the utility of the approach.

The probes and effectors are constitutively expressed throughout development. This is different from an inducible LoxP-Stop-LoxP approach. While clearly all the animals are viable, some discussion would be helpful that addresses how expression of a probe (for example a Ca indicator) might potentially alter function. Have the Authors assessed that this is not an issue experimentally?

---

## [Author Response]

Essential revisions:1) The manuscript would benefit from a transparent discussion of potential problems/limitations associated with the mice described here. For example, is there any evidence for off-target or incomplete expression of sensors or effectors? Are there any issues with the breeding of any of these lines?

We have included a new paragraph in the Discussion section addressing this issue (p. 29-30). Briefly, lines were screened for off-target expression and only those whose expression matched the previously reported expression pattern of the underlying gene locus were maintained. We did not observe any instances of off-site expression in the mouse lines that we report. However, while the expected general pattern of expression was matched in these lines, we do not assert absolute transcriptional fidelity. We now cite 3 examples of incomplete expression in the Discussion, although one of these (the HCN4^BAC^- CatCh line) is consistent with one report of HCN4 protein expression, and briefly address factors involved in transgene transcriptional control.

2) In at least one case, the description on the Jackson Laboratory website did not exactly match the description in the Major Resource Table. Jackson stock number 025406 links to a GCaMP3- mCherry mouse, but this is identified as GCaMP5-mCherry in the Table. Please carefully verify this vital information to avoid confusion.

We thank the reviewer. The Major Resources Table has been revised to include the Jackson Laboratory strain names (as assigned by Mouse Genome Informatics – MGI) in the last column for all strains.

3) As provided for experiments using endothelial optogenetic mice, controls using similar laser direction and parameters should be included for each experimental test of the optogenetic activators.

We have now included additional controls for optogenetic stimulation experiments (Figure 3—figure supplement 1, Figure 5—figure supplement 1, Figure 7—figure supplement 1).

4) A major advantage of using mice of the types described is the ease in which disease models, infection, additional genetic manipulations, environmental factors, aging, etc, can be superimposed upon the mice carrying the optogenetic activators and sensors. This point is mentioned only briefly in the discussion. The manuscript would be strengthened if an example of such an application could be provided in the manuscript. It would be nice to see a graded effect of optogenetic stimulation to provide more evidence of the utility of the approach.

We have expanded on this point in the new Discussion section and discussed an example of crossing CHROMus mice with disease models, although such experiments would be the subject of a separate, complete analysis. We also include a graded response to optogenetic stimulation (Figure 7C).

5) The probes and effectors are constitutively expressed throughout development. This is different from an inducible LoxP-Stop-LoxP approach. While clearly all the animals are viable, some discussion would be helpful that addresses how expression of a probe (for example a Ca indicator) might potentially alter function. Have the Authors assessed that this is not an issue experimentally?

We have included this point in the new Discussion section (p. 29-30).

Reviewer #1 (Recommendations for the authors):1. The manuscript would benefit from a transparent discussion of potential problems/limitations associated with the mice described here. For example, is there any evidence for off-target or incomplete expression of sensors or effectors? Are there any issues with the breeding of any of these lines?Anecdotally, the Acta2-GCaMP5-mCherry mice failed to breed in my laboratory, and I've heard from at least one other investigator who had the same issue. I have discussed this with some of the authors of the current study. Have any of the authors of the study encountered such problems with any of the other lines? It would be helpful to future users to disclose these issues and provide practical guidance.

We are aware of the Acta2-GCaMP5-mCherry (Acta2-GCaMP_GR) breeding problem and include information in the revised discussion.

2. In at least one case, the description on the Jackson Laboratory website did not exactly match the description in the Major Resource Table. Jackson stock number 025406 links to a GCaMP3- mCherry mouse, but this is identified as GCaMP5-mCherry in the Table. I didn't check them all. I suggest that the authors carefully verify this vital information to avoid confusion. Also, from the Jackson laboratory website, most, if not all, of these strains are cryopreserved. Recovery takes several months and is expensive. It might be helpful to potential users to disclose this limitation.

Thank you. The correct name of Acta2^BAC^-GCaMP-mCherry line is Acta2^BAC^-GCaMP_GR and we have corrected this and the entry for Cx40^BAC^-GCaMP_GR in Table 1 and Major Resources Table – thank you. The Major Resources Table has been revised to include the Jackson Laboratory strain names (as assigned by Mouse Genome Informatics – MGI) in the last column for all strains. We have now provided the reference for the GCaMP_GR reporter (Shui et al., 2014).

Reviewer #2 (Recommendations for the authors):The paper is well-written and thought provoking. The study is essentially a methods papers with some, if not all, of the mice described having been used in previous investigation-based studies.As provided for experiments using endothelial optogenetic mice, controls using similar laser direction and parameters should be given for each experimental test of the optogenetic activators.While the breeding tactics to generate experimentally useful chimeras are clear, it would also be helpful to compare relative expression/signal strengths for Cre recombinase driver mice vs. the monoallelic mice promoted in the present study. Comments about breeding success/difficulties would also be useful for those planning experiments using combinations of the mice from CHROMus.

We have expanded on this point in the discussion.

A major advantage of using mice of the types described is the ease in which disease models, infection, additional genetic manipulations, environmental factors, aging, etc, can be superimposed upon the mice carrying the optogenetic activators and sensors. This point is mentioned only briefly in the discussion, but frankly it is of such importance for future investigation that an example of such an application should be provided in the manuscript.

We appreciate this suggestion, but we feel that such a study, if done rigorously, should be the subject of a separate, focused analysis, yielding to specific disease insights, rather than an add on to this “toolbox” manuscript.

Reviewer #3 (Recommendations for the authors):I have a few suggestions:While the potential for this approach is vast and the data presented validates the approach, no new science or optogenetic/probes are really presented in this paper.It would be nice to see a graded effect of optogenetic stimulation to provide more evidence of the utility of the approach.

Thank you. This is now provided in Figure 7C.

The probes and effectors are constitutively expressed throughout development. This is different from an inducible LoxP-Stop-LoxP approach. While clearly all the animals are viable, some discussion would be helpful that addresses how expression of a probe (for example a Ca indicator) might potentially alter function. Have the Authors assessed that this is not an issue experimentally?

We now provide an expanded discussion of this point.